# Design for improving corrosion resistance of duplex stainless steels by wrapping inclusions with niobium armour

Shucai Zhang[1], Hao Feng[1], Huabing Li [1,2] ✉, Zhouhua Jiang[1], Tao Zhang[3], Hongchun Zhu[1], Yue Lin[1], Wei Zhang[4,5] & Guoping Li[6,7]

Unavoidable nonmetallic inclusions generated in the steelmaking process are fatal defects that often cause serious corrosion failure of steel, leading to catastrophic accidents and huge economic losses. Over the past decades, extensive efforts have been made to address this difficult issue, but none of them have succeeded. Here, we propose a strategy of wrapping deleterious inclusions with corrosion-resistant niobium armour (Z phase). After systematic theoretical screening, we introduce minor Nb into duplex stainless steels (DSSs) to form inclusion@Z core-shell structures, thus isolating the inclusions from corrosive environments. Additionally, both the Z phase and its surrounding matrix possess excellent corrosion resistance. Thus, this strategy effectively prevents corrosion caused by inclusions, thereby doubly improving the corrosion resistance of DSSs. Our strategy overcomes the long-standing problem of "corrosion failure caused by inclusions", and it is verified as a universal technique in a series of DSSs and industrial production.

Corrosion is one of the main causes of steel material failure, which not only leads to catastrophic safety and environmental accidents but also results in enormous economic losses[1,2]. Accordingly, a series of corrosion-resistant stainless steels (including austenitic, ferritic, and duplex steels, etc.) were designed to meet the requirements of long-life and stable service. However, these stainless steels still suffer serious corrosion failure when they serve in harsh environments with high chloride concentrations, high temperatures, and high pressures[3,4]. Unavoidable nonmetallic inclusions generated in the steelmaking process are one of the well-known inducers of corrosion failure, and the mechanisms triggering corrosion can be mainly classified into two categories: microdefects (element segregation, microcrevice formation, local stress concentration, etc.) surrounding the inclusions[5–7], and microgalvanic couples formed between conductive inclusions and the steel matrix[8,9]. In view of this difficult problem, deep deoxidization and desulfurization as well as inclusion modification treatments are often applied during steelmaking processes to alleviate the deleterious

effect of undesirable inclusions[10–14]. However, these methods are not very effective. The inclusion or surrounding matrix will still corrode, so the corrosion resistance improvement is very limited. To date, there is still no effective method to completely prevent the corrosion failure caused by inclusions. This has become a bottleneck problem in the long-term scientific and engineering practices of steel corrosion protection.

According to classical theory, nucleation is almost always heterogeneous in both liquids and solids[15], and inclusions can often serve as suitable nucleation sites[16,17]. Inspired by this phenomenon, can we precipitate a corrosion-resistant phase around an inclusion and wrap the inclusion through some treatment? Microalloying may be a feasible strategy because microalloying elements (Ti, V, Nb, etc.) in steel can easily combine with C and/or N to form precipitates such as carbides, nitrides, and carbonitrides[18–22]. If this strategy can be realized, then a certain armour-like precipitate will wrap the inclusion and isolate it from corrosive environments, thereby effectively preventing

[1]School of Metallurgy, Northeastern University, Shenyang 110819, China. [2]Key Laboratory for Ecological Metallurgy of Multimetallic Ores (Ministry of Education), Northeastern University, Shenyang 110819, China. [3]School of Materials Science and Engineering, Northeastern University, Shenyang 110819, China. [4]Central Iron and Steel Research Institute, Beijing 100081, China. [5]CITIC Metal Co., Ltd., Beijing 100027, China. [6]Shanxi Taigang Stainless Steel Co., Ltd., Taiyuan 030003, China. [7]State Key Laboratory of Advanced Stainless Steel Materials, Taiyuan 030003, China. ✉e-mail: lihb@smm.neu.edu.cn

localized corrosion. However, although Nb and Ti microalloying have certainly been applied in duplex stainless steels (DSSs), the relevant studies have mainly focused on their effects on the precipitation of Nb/Ti-bearing phases, chromium carbides and intermetallic phases, and the corresponding hot workability, mechanical properties and corrosion resistance[23–32]. In addition, some efforts have also been made to improve the corrosion resistance of DSSs by adding alloying elements such as Mo or W[33,34]. However, these methods cannot solve the localized corrosion problem induced by inclusions. To our knowledge, there is no report on wrapping inclusions with corrosion-resistant precipitates through the application of microalloying technology to improve the corrosion resistance of DSSs.

In this work, we propose a strategy to significantly improve the corrosion resistance of DSS by wrapping deleterious inclusions with corrosion-resistant niobium armour (Z phase). We first compare the feasibility of using Ti, V, and Nb elements for our strategy through systematic theoretical calculations and finally select Nb as the ideal microalloying element. As an example, S32205 DSS is microalloyed with minor Nb to implement this strategy. In the final prepared steel, the Nb-bearing Z phase indeed wraps the inclusions to form an inclusion@Z core-shell structure, thus isolating the inclusions from corrosive environments. Additionally, both the Z phase and its surrounding matrix possess excellent corrosion resistance. Therefore, this strategy effectively prevents corrosion caused by inclusions, doubly improving the corrosion resistance of S32205 DSS. Finally, we verify that our strategy can be universally applied to a series of DSSs as well as industrial production.

## Results

### Microalloying element selection

There are two key essential conditions for the implementation of our strategy: first, the precipitate can effectively wrap the inclusion; second, both the precipitate and its surrounding matrix have good corrosion resistance. To achieve the first essential condition, the following

three requirements must be met: (1) the precipitate should form after the inclusion, which is the prerequisite for wrapping the inclusion; (2) the lattice disregistry between the precipitate and inclusion should be low enough to realize heterogeneous nucleation of the precipitate around the inclusion[35]; and (3) the precipitate should stably exist instead of dissolving into the steel matrix during hot working and heat treatment procedures. To achieve the second essential condition, three other requirements must be met: (4) the precipitate should possess good corrosion resistance and not induce severe corrosion of the surrounding matrix[36]; (5) the potential difference between the precipitate and surrounding matrix should be small, and the potential of the precipitate should not be lower than that of the matrix to prevent the formation of a galvanic couple composed of a small anode and a large cathode, thus avoiding galvanic corrosion[11]; and (6) the precipitate should have a deformation capability compatible with the steel matrix to avoid microcrevices around the precipitate. When all the above conditions are met, stable and corrosion-resistant precipitates can wrap deleterious inclusions, thus preventing the corresponding localized corrosion. Therefore, selecting appropriate microalloying elements and precipitates is very important.

We explore the appropriate microalloying element and precipitate for S32205 DSS as an example. First, thermodynamic equilibrium calculations were performed by Thermo-Calc to analyze the precipitation behaviour of S32205 microalloyed with 0.25 wt.% Ti, V, or Nb. The chemical compositions used for calculations are given in Supplementary Table 1. The addition of Ti induces the formation of (Cr,Ti)N (Fig. 1a, d), whose initial precipitation temperature (1699 °C) exceeds even those of typical oxide (~1600 °C)[37] and sulfide (~1400 °C)[38] inclusions in steel, making wrapping of these inclusions almost impossible. Although the addition of V induces the formation of (Cr,V) N (Fig. 1b) that has a much lower initial precipitation temperature (1197 °C) than the inclusions, the Cr content in (Cr,V)N is close to 70 wt.% (Fig. 1e), so its formation will induce severe Cr depletion in the surrounding matrix, which deteriorates the corrosion resistance of

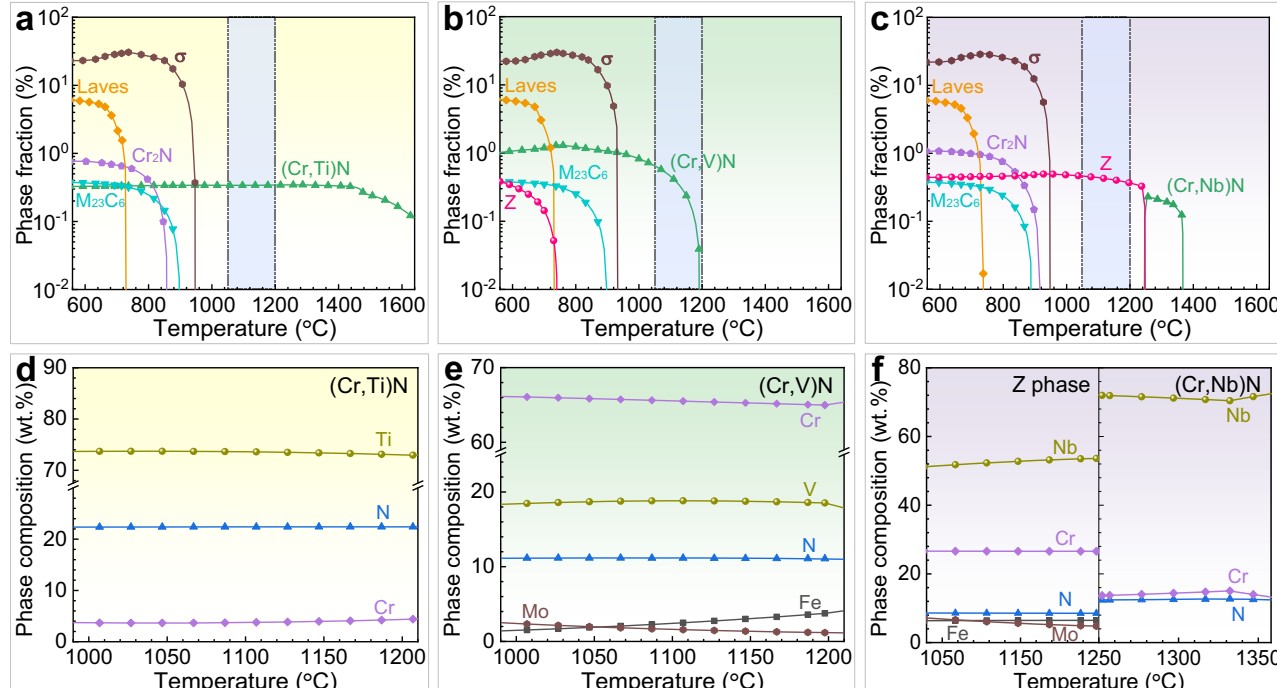

**Fig. 1 | Thermo-Calc calculation results of microalloyed S32205 DSSs.** Precipitation behaviour of phases in S32205 DSSs microalloyed with **a** 0.25 wt.% Ti, **b** 0.25 wt.% V, and **c** 0.25 wt.% Nb. The light blue shaded area represents the approximate temperature range of hot working and heat treatment. Chemical compositions of **d** (Cr,Ti)N, **e** (Cr,V)N, and **f** Z phase and (Cr,Nb)N in (**a**), (**b**), and (**c**), respectively. (Cr,Ti)N, (Cr,V)N, and (Cr,Nb)N belong to MN-type precipitates whose definitions are based on their calculated compositions in (**d**), (**e**), and (**f**), respectively. Source data are provided as a Source Data file.

steel[39,40]. By comparison, the addition of Nb promotes the formation of (Cr,Nb)N, whose initial precipitation temperature (1370 °C) is also lower than that of the inclusions[37,38]. At ~1250 °C, the (Cr,Nb)N phase completely transforms into the Z phase which can stably exist during hot working and heat treatment procedures (as indicated by the light blue shaded area in Fig. 1c). Furthermore, the (Cr,Nb)N and Z phases exhibit very similar precipitation and transformation behaviours within the composition range of key elements (Cr, Mo, N, and Nb) of steel (Supplementary Fig. 1). Thus, in the final product, a structure of the Z phase wrapped around the inclusion is very likely to be formed. Interestingly, the Z phase mainly contains ~50 wt.% Nb and a certain amount of Cr, Mo, and N (Fig. 1f). In particular, its Cr content is close to that of the steel matrix. This indicates that the Z phase not only is a corrosion-resistant precipitate but also does not induce Cr depletion in the matrix. Therefore, Nb microalloying shows considerable promise in terms of wrapping inclusions with a Nb-bearing phase without inducing corrosion of itself or matrix.

To evaluate the effectiveness of heterogeneous nucleation between the Nb-bearing phase and inclusions, the lattice disregistries were calculated according to the Bramfitt two-dimensional disregistry model[35,41]. (Cr,Nb)N belongs to a NbN-type precipitate, which has a face-centred cubic structure ($a$ = 0.439 nm), while the Z phase has a tetragonal structure ($a$ = 0.3037 nm and $c$ = 0.7391 nm). As listed in Supplementary Table 2, the lattice disregistries between the (Cr,Nb)N

or Z phase and typical inclusions ($MgAl_2O_4$ and MnS) are much lower than the coherency criterion of 12%[35], indicating that both the (Cr,Nb)N and Z phases can theoretically heterogeneously nucleate around these inclusions and then wrap them. Taking the above calculation results into account, we selected Nb microalloying as the ideal solution to wrap inclusions, thereby improving the corrosion resistance of the steel.

## Material processing and microstructure characterization

To verify our proposal, three S32205 DSSs with various Nb contents (0 Nb, 0.10 Nb, and 0.25 Nb) were manufactured using vacuum induction melting under a nitrogen atmosphere, and their compositions are listed in Supplementary Table 3. Due to the adoption of the same raw materials and melting procedures, the inclusions in the three steels are similar in size, number density, and type ($MgAl_2O_4$, MnS, and $MgAl_2O_4$-MnS, Supplementary Fig. 2). Additionally, the three steels after hot working and heat treatment have ferrite-austenite duplex microstructures, and the addition of Nb leads to a slight increase in the ferrite phase fraction because Nb is a ferrite former and can take a few N atoms away from austenite (Supplementary Fig. 3). The micromorphologies show that a small amount of the Z phases form in 0.10 Nb steel and wrap some inclusions (Fig. 2b). In 0.25 Nb steel, a substantial portion of inclusions are completely wrapped by the Z phase (Fig. 2c). In other words, minor Nb addition indeed promotes the

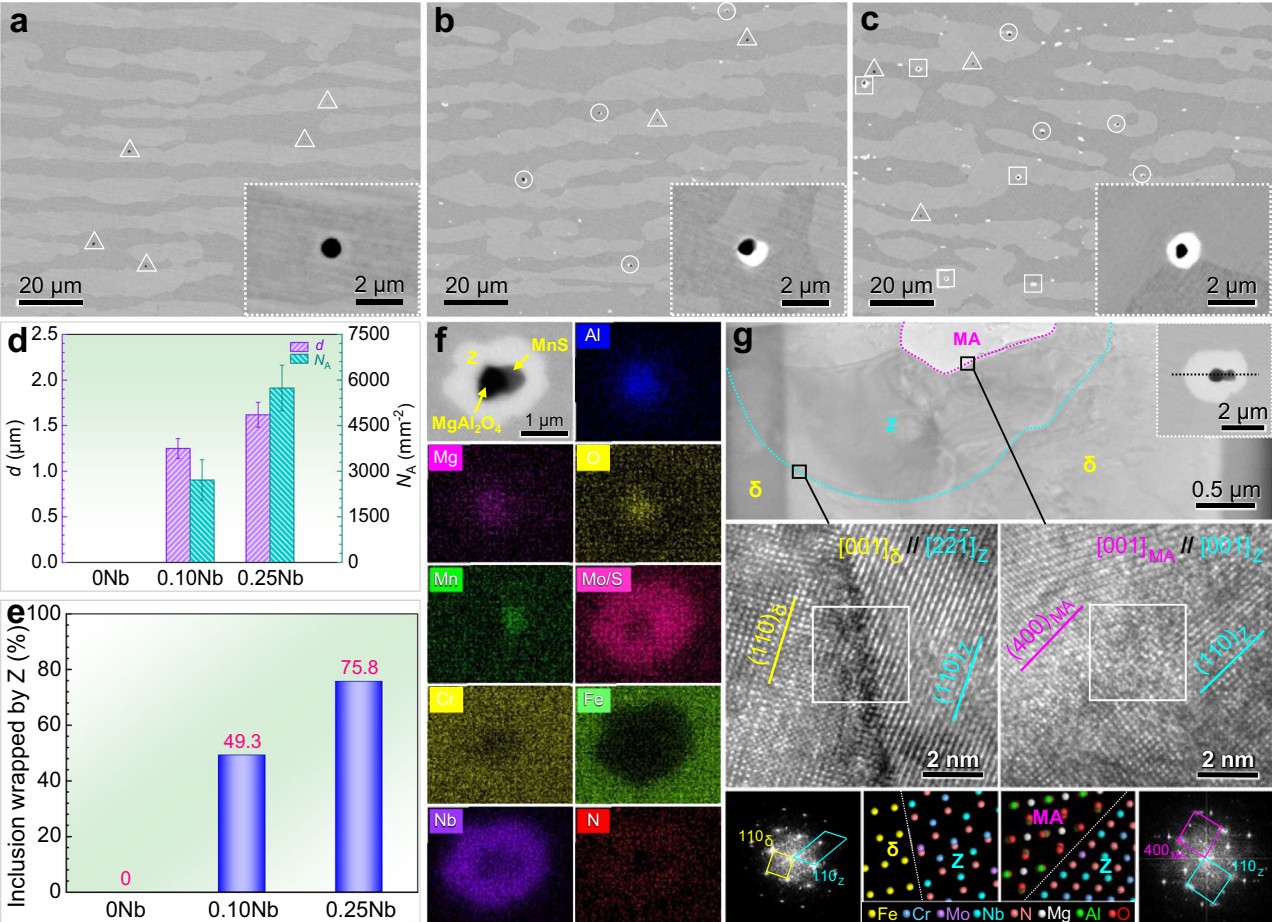

**Fig. 2 | Characterization of the inclusion@Z core-shell structure.** Occurrence state of the inclusion and the Z phase in solution-treated S32205 DSSs: **a** 0 Nb, **b** 0.10 Nb, and **c** 0.25 Nb. The triangle, circle, and square indicate the single inclusion, the inclusion wrapped by the Z phase partially and completely, respectively. The inserts show the morphologies of single inclusion and the inclusion@Z core-shell structure. **d** Average equivalent diameter ($d$) and number density ($N_A$) of

the Z phase. All the error bars in (**d**) represent the standard deviation ($n$ = 30 independent experiments). **e** Proportion of inclusions wrapped by the Z phase. **f** EDS elemental mappings of the inclusion@Z core-shell structure in 0.25 Nb steel. **g** TEM, HRTEM images and associated diffraction patterns of the inclusion@Z core-shell structure in 0.25 Nb steel. MA in (**g**) is the abbreviation for the $MgAl_2O_4$ inclusion, and δ in (**g**) is ferrite phase. Source data are provided as a Source Data file.

formation of the Z phase around inclusions, ultimately forming an inclusion@Z core-shell structure. Combining the statistical results in Fig. 2d, e, the size and number density of the Z phase as well as the proportion of inclusions wrapped by the Z phase all increase with increasing Nb content. The elemental mappings of the core-shell structure further illustrate that the core inclusion is fully wrapped by the external (Nb, Cr, Mo, N)-bearing Z phase (Fig. 2f and Supplementary Fig. 4). Additionally, no Cr-depleted zone and only slightly Mo- and N-depleted zones were detected near the Z phase (Supplementary Fig. 5), and their effect on the corrosion resistance will be discussed in the next section.

Moreover, the morphology (Fig. 2g) along the longitudinal section of the inclusion@Z core-shell structure further verifies that the inclusion is completely wrapped by the Z phase and that no microcrevice forms around the Z phase. High-resolution transmission electron microscopy (HRTEM) images and fast Fourier transform (FFT) patterns suggest that the $[001]_Z$ zone axis is parallel to the $[001]_{MgAl2O4}$ zone axis. Additionally, the $(110)_Z$ and $(1\bar{1}0)_Z$ planes are approximately parallel to $(400)_{MgAl2O4}$ and $(040)_{MgAl2O4}$ planes, respectively. Therefore, the Z phase and $MgAl_2O_4$ inclusion exhibit the following specific orientation relationship (OR): $(110)[001]_Z \sim //(400)[001]_{MgAl2O4}$ and $(1\bar{1}0)[001]_Z \sim //(040)[001]_{MgAl2O4}$. Similarly, the OR between the Z phase and the ferrite ($\delta$) substrate is confirmed to be $(110)[2\bar{2}\bar{1}]_Z \sim //(110)[001]_\delta$ and $(102)[2\bar{2}\bar{1}]_Z \sim //(1\bar{1}0)[001]_\delta$. The HRTEM images also indicate that the $Z/MgAl_2O_4$ and $Z/\delta$ interfaces are almost straight, which is conducive to reducing the interfacial energy[42]. In general,

both the lattice misfit and interfacial energy can reflect the effectiveness of heterogeneous nucleation between a nucleated phase and a substrate[43,44]. Thus, the lattice misfit and interfacial energy of the $Z/MgAl_2O_4$ and $Z/\delta$ interfaces were calculated according to the above interfacial structure parameters[15,45]. The lattice misfits of the $Z/MgAl_2O_4$ and $Z/\delta$ interfaces are 8.1% and 3.5%, and the corresponding interfacial energies are 0.419 and 0.349 J·m$^{-2}$, respectively (Supplementary Table 4). Both interfacial energies are lower than the critical value of 0.6 J·m$^{-2}$, indicating that both the $Z/MgAl_2O_4$ and $Z/\delta$ interfaces are semicoherent[46]. These results further reveal that the formation of the Z phase around $MgAl_2O_4$ is highly effective, and the Z phase shows good lattice matching with the $\delta$ substrate.

## Corrosion behaviour

To assess the corrosion resistance, we carried out electrochemical and immersion corrosion tests on S32205 DSSs with various Nb contents. Figure 3a shows the potentiodynamic polarization curves in double-concentration simulated seawater at 72 °C (pH 8.2), which can simulate the harshest corrosion conditions in the low-temperature multistage flash evaporation process of seawater desalination equipment[47]. The pitting potentials of the Nb-bearing steels are noticeably higher than that of the Nb-free steel. Moreover, the addition of Nb dramatically weakens the current fluctuations in the passive ranges, reducing the metastable pitting susceptibility[48]. The cumulative probability distribution of the pitting potential (Supplementary Fig. 6a) further reveals that increasing the Nb content to 0.25 wt.% doubly enhances

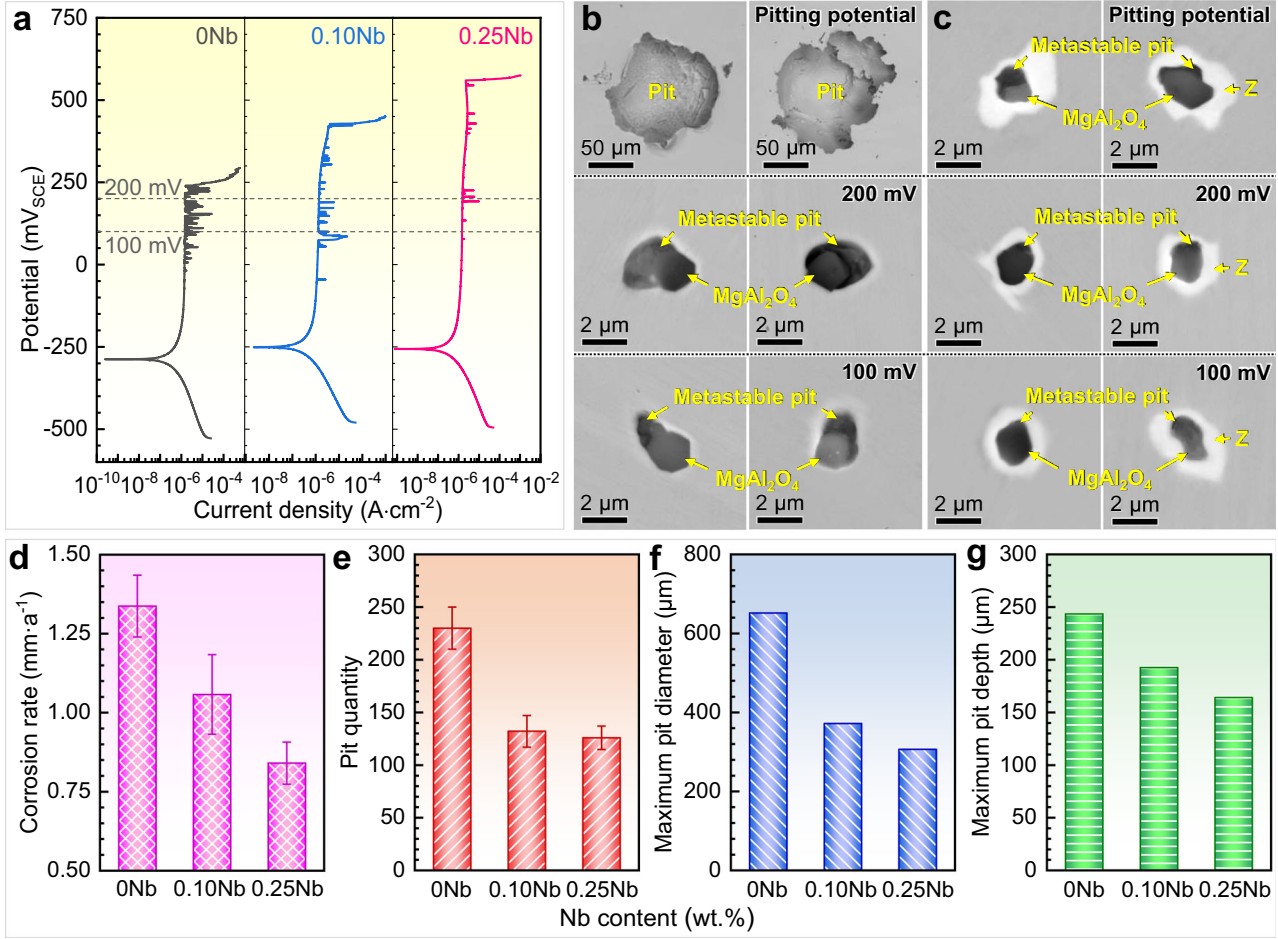

**Fig. 3 | Electrochemical and immersion corrosion behaviour of S32205 DSSs.** Electrochemical corrosion behaviour in double-concentration simulated seawater at 72 °C (pH 8.2): **a** potentiodynamic polarization curves and corrosion morphologies of **b** 0 Nb and **c** 0.25 Nb steels. Immersion corrosion behaviour in a 6% FeCl₃ solution at 50 °C for 12 h: **d** corrosion rate, **e** pit quantity on the sample surface, **f** maximum pit diameter, and **g** maximum pit depth. All the error bars in (**d**) and (**e**) represent the standard deviation ($n = 3$ independent experiments). Source data are provided as a Source Data file.

the pitting potential of S32205 DSSs. Figure 3b, c shows the corrosion morphologies of inclusions and the inclusion@Z core-shell structure after potentiodynamic polarization tests stopped at 100 mV, 200 mV, and the pitting potential. Visible metastable pits formed around the individual inclusions in 0 Nb steel at 100 mV and 200 mV and further expanded into large corrosion pits at the pitting potential. For the inclusion@Z core-shell structure in 0.25 Nb steel, although metastable pits also formed around the core inclusions at low potentials, they did not significantly expand at the pitting potential. These phenomena indicate that the Z phase is indeed very corrosion resistant, so local acidification in the pits caused by inclusion dissolution will not induce apparent corrosion of the Z phase. Furthermore, the corrosion-resistant Z phase shell effectively prevents the development of metastable pits into stable pits. Meanwhile, the matrix surrounding the Z phase was also not corroded after potentiodynamic polarization testes, indicating that it still had good corrosion resistance in simulated seawater. Supplementary Fig. 7a shows the cyclic polarization curves in simulated seawater, which indicate that the Nb-bearing steels exhibited very similar protective potentials (repassivation ability) to the Nb-free steel. As schematically shown in Supplementary Fig. 7b, neither the Z phase nor its surrounding matrix in Nb-bearing steel was corroded. Thus, corrosion mainly occurred at the steel matrix with defects. Accordingly, the repassivation also occurred at the front of the corroded steel matrix. Therefore, both Nb-bearing and Nb-free steels underwent repassivation in the steel matrix, so they exhibited similar repassivation behaviours.

In addition, the higher corrosion resistance of the Nb-bearing steels was further validated by immersion corrosion tests conducted in a 6% FeCl$_3$ solution at 50 °C for 12 h (Fig. 3d–g). The addition of Nb leads to reductions in the corrosion rate as well as the quantity, maximum diameter, and maximum depth of the pit cavities. Meanwhile, the Gumbel extreme value distribution was used to explore the effect of Nb microalloying on pit growth (Supplementary Note 1 and Supplementary Fig. 8). The results show that the pit growth of three S32205 DSSs can be well modelled by the Gumbel extreme value distribution. Once the inclusions are wrapped by the Z phase, the proportion of inclusions that may induce deep pits is considerably decreased; therefore, the maximum pit depth of Nb-bearing steels becomes significantly smaller. In summary, both the electrochemical and immersion corrosion results confirm that Nb addition significantly improves the corrosion resistance of S32205 DSSs.

Generally, the factors affecting corrosion resistance mainly include the passive film composition, defects around inclusions or precipitates, microgalvanic couples between inclusions or precipitates and the steel matrix, etc[5–9]. To clarify the relevant mechanisms in this work, we first analyzed the influence of Nb on the passive film composition (Supplementary Fig. 9). There are no significant differences in the O and Cr concentrations or Cr-bearing components of these steels. Additionally, there is no apparent enrichment of Nb-bearing oxides in the passive film. Thus, it can be concluded that the addition of Nb has little influence on the passive film composition. To assess the wrapping effect on the corrosion process, we performed ex situ observations and potential distribution analysis of the MgAl$_2$O$_4$-MnS inclusions and MgAl$_2$O$_4$-MnS@Z core-shell structure before and after immersion corrosion tests conducted in 6% FeCl$_3$ solution at 50 °C (Fig. 4). After immersion for 12 h, the MnS part of the MgAl$_2$O$_4$-MnS composite inclusions in 0 Nb steel rapidly dissolved and induced apparent pitting corrosion (Fig. 4a, b). This occurred because the MnS inclusion possesses a much lower Volta potential (i.e., a higher electrochemical activity[49]) than the steel matrix (Fig. 4f, g), which easily induces galvanic corrosion. Accordingly, the MnS inclusion serves as an anode and preferentially dissolves, while the steel matrix serves as a cathode, accelerating the development of pitting corrosion[50]. In addition, the microcrevice around MgAl$_2$O$_4$ in 0 Nb steel also induced apparent crevice corrosion (Supplementary Fig. 10). After immersion for 10 d,

the core MnS part of the MgAl$_2$O$_4$-MnS@Z core-shell structure in 0.25 Nb steel dissolved, and some regions of the steel matrix near and far from the Z phase were also corroded. However, no visible corrosion sign was found on the Z phase (Fig. 4c, d and Supplementary Fig. 11). These facts indicate that the corrosion resistance of the Z phase is much higher than that of the steel matrix and inclusions, and local acidification in the pits will not induce corrosion of the Z phase. This is because the (Nb, Cr, Mo, N)-rich Z phase is a corrosion-resistant precipitate with a Volta potential (i.e., electrochemical activity[11]) similar to that of the steel matrix (Fig. 4f, g), and no microcrevice forms around the Z phase. Notably, in highly aggressive 6% FeCl$_3$ solution, the steel matrix far from the Z phase has already corroded, so the corrosion of the regions surrounding the Z phase should be acceptable.

In addition, the double loop electrochemical potentiokinetic reactivation (DL-EPR) results (Supplementary Fig. 12) show that the Nb-bearing steels exhibit similar degrees of sensitization to the Nb-free steel, indicating that the slight Mo and N depletion surrounding the Z phase has a negligible effect on the intergranular corrosion (IGC) resistance of the Nb-bearing steels. Apparently, corrosion mainly occurred within the γ phase and at the γ/δ boundaries (Supplementary Fig. 13). Although the depleted zones around the Z phase were also corroded, this corrosion was negligible compared to the extensive corrosion of the γ phase and the γ/δ boundaries because the total area of the depleted zones was much smaller than that of the γ phase and the γ/δ boundaries.

In summary, wrapping inclusions with the Z phase not only can prevent galvanic or microcrevice corrosion caused by inclusions but also does not induce corrosion of the matrix surrounding the Z phase. Meanwhile, this strategy will not reduce the IGC resistance of the steel. In other words, the Z phase is similar to a layer of corrosion-resistant niobium armour wrapped around the inclusions, effectively preventing corrosion failure.

## Universality of Nb microalloying technology

To assess the universality of the Nb microalloying technology in a series of DSSs, the precipitation behaviours of several DSSs (including lean S32101 and S32304, standard S32205, super S32750, and hyper S32707) microalloyed with 0.25 wt.% Nb were calculated by Thermo-Calc software (Fig. 5a). The chemical compositions of these DSSs used for calculations are given in Supplementary Table 5. The results show that the Z phase can form in all types of Nb microalloyed DSSs through (Cr,Nb)N transformation or direct precipitation, and the initial formation temperatures (1318–1378 °C) are much lower than those of inclusions[37,38]. Accordingly, the Z phase is also greatly promising for wrapping inclusions in these DSSs. To verify this conjecture, we selected two representative DSSs (lean S32101 and super S32750) and carried out manufacturing, microstructure observation, and potentiodynamic polarization tests, all of which were similar to those for S32205 DSS. The chemical compositions of the experimental S32101 and S32750 DSSs are given in Supplementary Table 6. When both types of steels are microalloyed with 0.25 wt.% Nb, the Z phase indeed forms around the inclusions and wraps them (inserts in Fig. 5b, c). Accordingly, the pitting potentials of both Nb-bearing steels are evidently enhanced, as shown in the potentiodynamic polarization curves (Fig. 5b, c) and cumulative probability distribution of the pitting potential (Supplementary Fig. 6b, c). Additionally, we also applied our strategy in a stainless steel factory. Practice proves that our strategy can also wrap inclusions and significantly improve the corrosion resistance of industrial S32205 DSS produced by continuous casting and hot continuous rolling (Fig. 5d and Supplementary Fig. 6d). Meanwhile, the Nb-bearing steels exhibit an apparent increase in strength and a marginal decrease in ductility compared with the Nb-free steel (Supplementary Table 7). In summary, Nb microalloying is a widely applicable strategy for significantly improving the corrosion resistance of various DSSs by wrapping deleterious inclusions with niobium armour (Z phase).

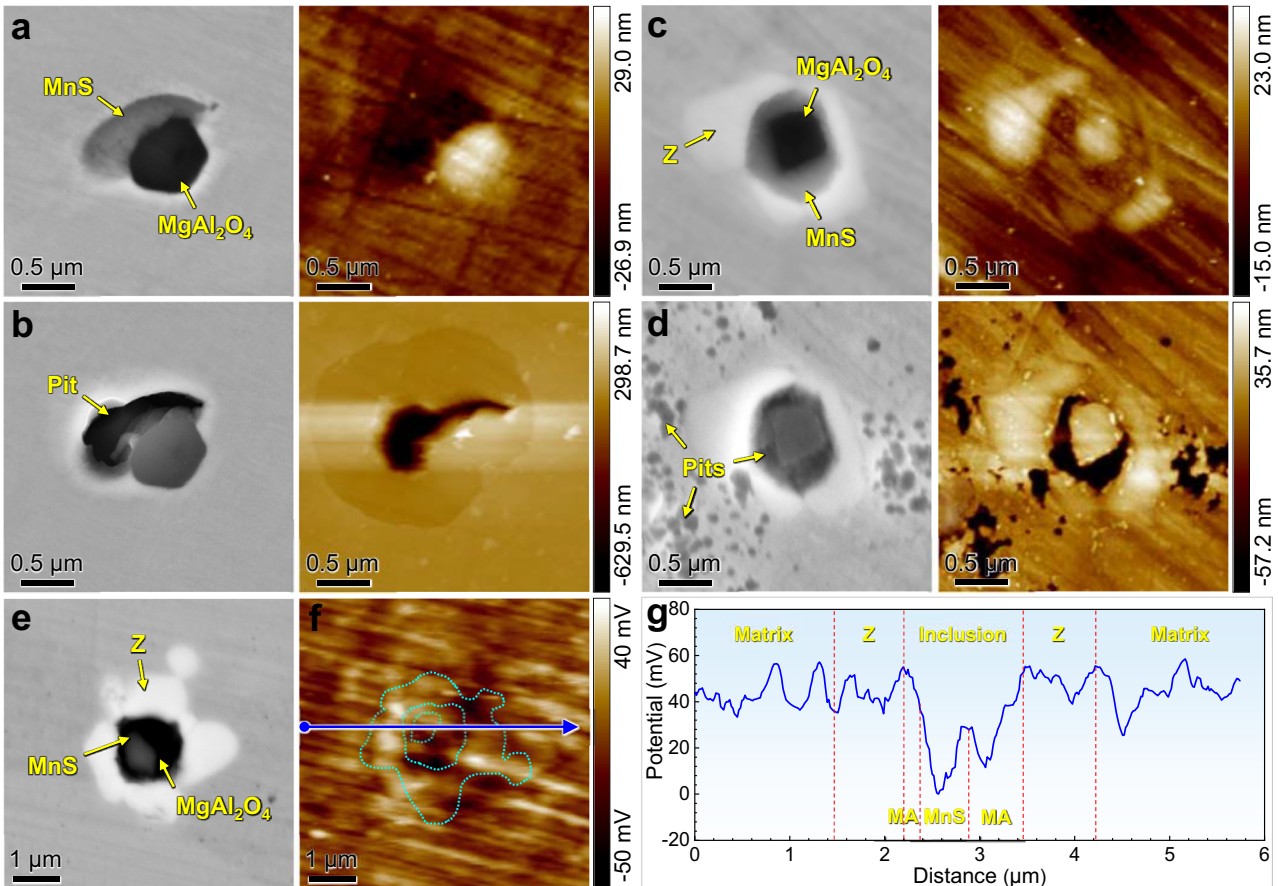

**Fig. 4 | Characterization of S32205 DSSs before and after immersion corrosion.** SEM morphologies (left) and AFM topographies (right) before (**a**, **c**) and after (**b**, **d**) immersion corrosion in a 6% FeCl$_3$ solution at 50 °C for **b** 12 h and **d** 10 d. **a**, **b** MgAl$_2$O$_4$-MnS inclusion in 0 Nb steel and **c**, **d** MgAl$_2$O$_4$-MnS@Z core-shell structure in 0.25 Nb steel. **e** SEM morphology and **f** SKPFM map of MgAl$_2$O$_4$- MnS@Z core-shell structure before immersion corrosion. **g** Volta potential variation along the MgAl$_2$O$_4$-MnS@Z core-shell structure as marked by the arrowed line in (**f**). MA in (**g**) is the abbreviation for the MgAl$_2$O$_4$ inclusion. Source data are provided as a Source Data file.

## Discussion

Corrosion failure caused by inclusions has been a bottleneck problem in long-term scientific and engineering practices. Accordingly, we proposed a strategy of wrapping deleterious inclusions with corrosion-resistant niobium armour (Z phase). The successful application of our strategy mainly depends on two key essential conditions: the first is employing Nb microalloying to form an inclusion@Z core-shell structure, thus isolating the inclusions from corrosive environments; the second is ensuring that the Z phase and its surrounding matrix possess good corrosion resistance. We will discuss in detail below how to achieve these two essential conditions.

To achieve the first essential condition of our strategy, ensuring that the Z phase can form around inclusions and then wrap them is necessary. According to the nonequilibrium solidification process (Supplementary Fig. 14), MgAl$_2$O$_4$ inclusion is expected to first be generated from the liquid steel, which can stably exist and easily act as heterogeneous nucleation core for later formed phases. During the subsequent solidification process, the ferrite (δ) phase, MnS inclusion, and austenite (γ) phase start to precipitate from the liquid steel at -1444, 1418, and 1305 °C, respectively. Reportedly[51,52], MnS is prone to nucleating around preexisting MgAl$_2$O$_4$ to form MgAl$_2$O$_4$-MnS composite inclusions. As solidification continues, some alloy elements are gradually enriched in the residual liquid phase, leading to apparent positive segregation[53]. At -1295 °C, the enrichment of Nb, Cr, and N promotes the precipitation of (Cr,Nb)N. Because the lattice disregistries between (Cr,Nb)N and the main inclusions (MgAl$_2$O$_4$ and

MnS) are much lower than the coherency criterion of 12%[35] (Supplementary Table 2), (Cr,Nb)N can easily nucleate around these inclusions, thereby forming MgAl$_2$O$_4$@(Cr,Nb)N, MnS@(Cr,Nb)N, and MgAl$_2$O$_4$-MnS@(Cr,Nb)N core-shell structures (Supplementary Fig. 15).

Before hot working, the cast ingot experienced a heat treatment at 1180 °C for 1 h. Interestingly, the (Cr,Nb)N phase completely transformed into the Z phase during the heating process (Supplementary Fig. 16). This phase transformation is mainly achieved through the continuous replacement of Nb in (Cr,Nb)N by Cr and Mo in the surrounding matrix, due to the strong chemical driving force between them. Accordingly, the inclusion@(Cr,Nb)N core-shell structure also transformed into an inclusion@Z core-shell structure. Additionally, during the subsequent hot working and heat treatment procedures, additional Z phases can nucleate around the inclusions owing to the low lattice disregistries between them (Supplementary Table 2). As verified for an example, the lattice misfit and interfacial energy of the Z/MgAl$_2$O$_4$ interface are very low (Supplementary Table 4). On the one hand, the low lattice misfit reflects that few mismatch dislocations and low lattice distortion are generated at the Z/MgAl$_2$O$_4$ interface, which can better accommodate mismatch strain[54]. Thus, the atomic crystal planes of the Z phase and MgAl$_2$O$_4$ well match at the interface, indicating easy heterogeneous nucleation between them. On the other hand, the low interfacial energy reveals that the energy barrier for the nucleation of the Z phase around MgAl$_2$O$_4$ is very low[55]. Namely, the critical conditions for heterogeneous nucleation can be easily satisfied,

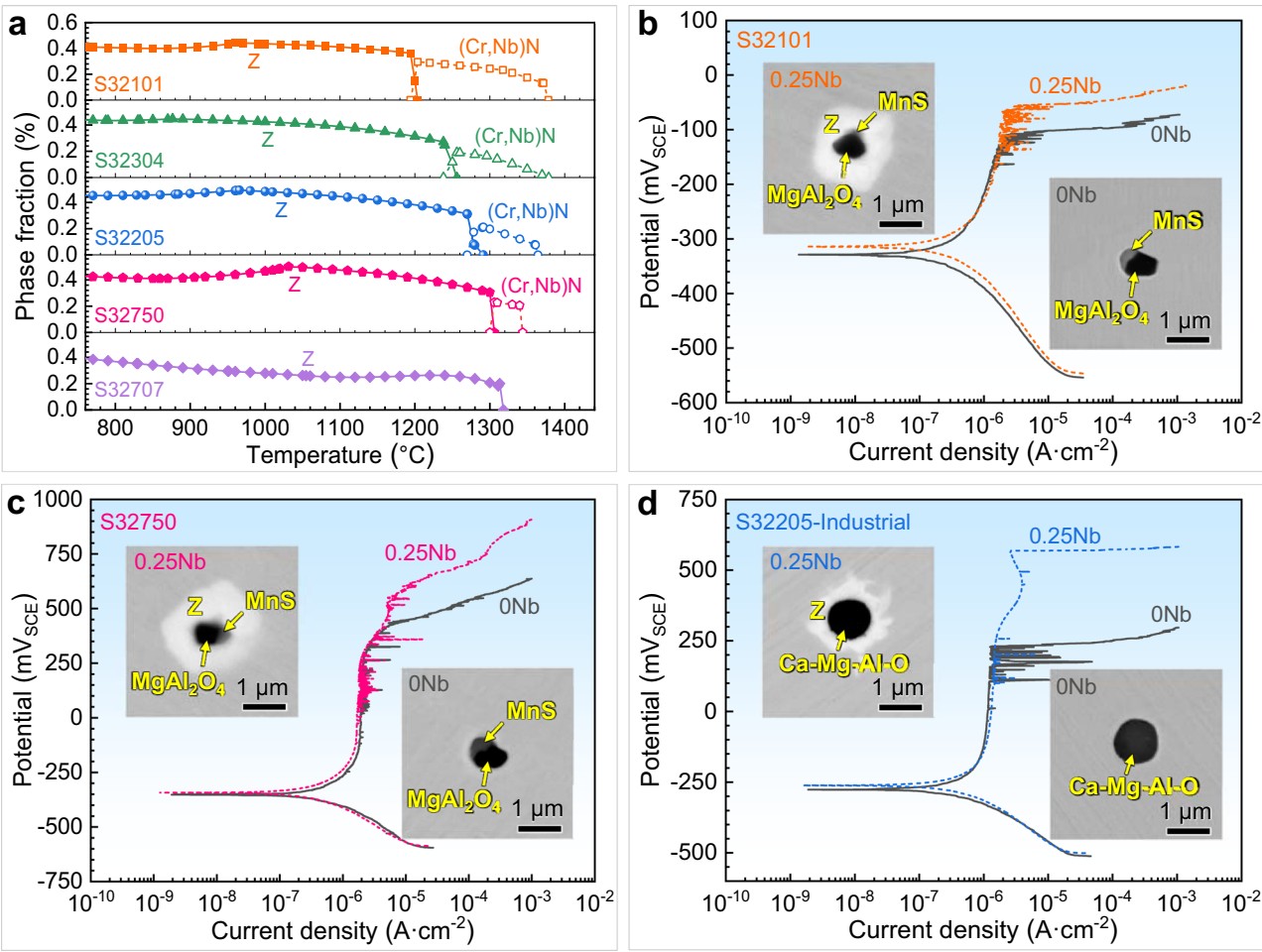

**Fig. 5 | Universality of Nb microalloying technology. a** Precipitation behaviour of Nb-bearing phases in series of DSSs microalloyed with 0.25 wt.% Nb, which were calculated by Thermo-Calc software. Potentiodynamic polarization curves of **b** S32101, **c** S32750, and **d** industrial S32205 DSSs microalloyed with and without 0.25 wt.% Nb in double-concentration simulated seawater at 72 °C (pH 8.2). The inserts show the morphologies of single inclusion and inclusion@Z core-shell structure. Source data are provided as a Source Data file.

triggering Z phase nucleation and enhancing the interfacial stability. Therefore, the strain-induced precipitation effect during hot working[56] and the isothermal aging effect during heat treatment[57] further promote the precipitation of the Z phase around inclusions to form more inclusion@Z core-shell structures. Accordingly, the proportion of inclusions wrapped by the Z phase in 0.25 Nb S32205 DSS increases from 51.8% (before hot working) to 75.8% (after hot working and heat treatment, Fig. 2e). Consequently, the first essential condition of our strategy (i.e., wrapping deleterious inclusions with the Nb-bearing Z phase) is achieved.

To achieve the second essential condition of our strategy, ensuring that the Z phase is a corrosion-resistant phase and will not induce severe corrosion of the surrounding matrix is necessary. In general, the main factors affecting the corrosion resistance of a precipitate and its surrounding matrix are the differences between their compositions, potentials, and plasticities[58-60]. Both the calculated chemical composition (Fig. 1f) and elemental mappings (Fig. 2f) show that the Z phase is a (Nb, Cr, Mo, N)-bearing precipitate, suggesting that it is indeed highly corrosion resistant. Moreover, although slightly Mo- and N-depleted zones were detected near the Z phase, these regions still contained relatively high contents of Cr, Mo, and N elements (Supplementary Fig. 5), so they still had good corrosion resistance in simulated seawater. According to electrochemical theory[61], two adjacent phases with large potential difference will induce galvanic corrosion. In particular, the greater the potential difference between them is, the more serious the galvanic corrosion. In the present work, the electrode potential of the Z phase is basically the same as that of the surrounding matrix (Fig. 4f, g), so they will not form a corrosive galvanic couple composed of a small anode and a large cathode. Namely, no galvanic corrosion occurs between them. Therefore, neither the Z phase nor the surrounding matrix easily dissolves in corrosive environments. In addition, microcrevices may form at the precipitate/matrix interface due to their different plasticities[60], which further induces microcrevice corrosion[62]. However, no microcrevice forms at the interface of the Z phase and matrix (Fig. 2f, g), which is attributed to their basically similar nanohardness and elastic modulus values (Supplementary Table 8). Thus, no microcrevice corrosion occurs around the Z phase. Based on the above analyses, we firmly believe that both the Z phase and its surrounding matrix possess good corrosion resistance. The morphologies after electrochemical corrosion tests (Fig. 3c) further confirmed that no visible corrosion sign was found on the Z phase and its surrounding matrix, and local acidification in the pits caused by inclusion dissolution did not induce corrosion of the Z phase. Although parts of the matrix surrounding the Z phase were corroded after long-term immersion in highly aggressive 6% FeCl₃ solution, the steel matrix far from the Z phase was also corroded (Fig. 4d and Supplementary Fig. 11). Therefore, this universal corrosion in extremely harsh environments is acceptable. Additionally, the slight Mo and N depletion around the Z phase has a negligible influence on the repassivation behaviour (Supplementary Fig. 7) and

IGC resistance of the steel (Supplementary Figs. 12 and 13). Accordingly, the second essential condition of our strategy is also realized. To sum up, wrapping deleterious inclusions with the Z phase can effectively isolate them from corrosive environments, significantly improving the corrosion resistance of DSSs.

In summary, our strategy of wrapping deleterious inclusions with niobium armour (Z phase) overcomes the long-standing problem of "corrosion failure caused by inclusions", thereby significantly improving the corrosion resistance of DSSs. This technique was verified to be universal in a series of DSSs as well as in industrial production. Clearly, this strategy paves a pathway for corrosion protection of stainless steels and represents progress in ensuring long-life and safe operation of high-end equipment constructed by DSSs.

## Methods

### Theoretical calculations

For the primary selection of microalloying elements and universality verification of Nb microalloying technology, the precipitation behaviours of various DSSs were calculated using Thermo-Calc software with the TCFE9 database. The chemical compositions of DSSs used for calculations are listed in Supplementary Tables 1 and 5. The Gulliver-Scheil model in FactSage software was applied to predict the nonequilibrium solidification process of 0.25 Nb S32205 DSS based on the chemical composition in Supplementary Table 3. To assess the effectiveness of heterogeneous nucleation between Nb-bearing phases and inclusions, their lattice disregistries were calculated according to the Bramfitt two-dimensional disregistry model (Eq. (1))[35,41]. Furthermore, the lattice misfit and interfacial energy between the Z phase and inclusions or ferrite were calculated using Eqs. (2)–(6) [15,45] based on the interfacial structure parameters determined by HRTEM.

$$\delta = \sum_{i=1}^{3} \left| \frac{d_{[uvw]_s^i} \cos\theta - d_{[uvw]_n^i}}{3 \times d_{[uvw]_n^i}} \right| \times 100 \tag{1}$$

where $\delta$ (%) is the lattice disregistry; $[uvw]_s$ and $[uvw]_n$ are the low-index directions of the substrate and the nucleated phase, respectively; $d_{[uvw]_s}$ (nm) and $d_{[uvw]_n}$ (nm) are the interatomic spacings along $[uvw]_s$ and $[uvw]_n$, respectively; and $\theta$ (°) is the angle between $[uvw]_s$ and $[uvw]_n$.

$$\delta' = \left| \frac{d_{(hkl)_s} - n d_{(hkl_n)} \cos a}{d_{(hkl)_n}} \right| \times 100 \tag{2}$$

$$\sigma_1 = \frac{Gb_1}{4\pi(1-\nu)} f(\delta_1') \tag{3}$$

$$\sigma_2 = \frac{Gb_2}{4\pi(1-\nu)} f(\delta_2') \tag{4}$$

$$f(\delta') = \delta' \left[ \frac{2}{1 + \frac{1}{4\delta'^2}} - \ln(2\delta') \right] \tag{5}$$

$$\bar{\sigma} = \frac{2}{\frac{1}{\sigma_1} + \frac{1}{\sigma_2}} \tag{6}$$

where $\delta'$ (%) is the lattice misfit of matching planes; $d_{(hkl)_s}$ (nm) and $d_{(hkl)_n}$ (nm) are the interatomic spacings along the $(hkl)_s$ and $(hkl)_n$ planes, respectively; $n$ is the corresponding period of crystallographic planes; $\alpha$ (°) is the angle between the $(hkl)_s$ and $(hkl)_n$ planes; $G$ (106.8 GPa) is the shear modulus of the Z phase at 1250 °C, calculated by JMatPro software; $b$ is the corresponding Burgers vector of matching

planes; $\nu$ is Poisson's ratio (0.26)[63]; and $\sigma$ and $\bar{\sigma}$ are interfacial energy and average interfacial energy of matching planes, respectively.

### Material preparation

S32101, S32205, and S32750 DSSs with various Nb contents were manufactured using vacuum induction melting under nitrogen atmosphere. During the melting process, small amounts of metallic aluminium and nickel-magnesium alloy were successively added for deoxidization. The chemical compositions of these experimental DSSs are listed in Supplementary Tables 3 and 6. The ingots were first kept at 1180 °C for 1 h and then hot forged and hot rolled into 12 mm plates in the temperature range of 950 –1180 °C. To control the appropriate grain size and phase ratio, these plates were solution treated at 1050 °C (S32101 and S32205 DSSs) and 1100 °C (S32750 DSS) for 0.5 h, followed by water quenching.

### Microstructure characterization

Several samples were cut from the solution-treated plates, ground with 2000 grit silicon carbide paper and polished with 1.5 μm diamond paste. The samples for metallographic observation were electrolytically etched at 6–8 V for 10–20 s in 30% KOH solution. The ferrite and austenite phase fractions were measured using at least 30 metallographic micrographs collected by an optical digital microscope (ODM, Olympus DSX 510). The ferrite-austenite duplex microstructure was examined by electron backscatter diffraction (EBSD) with a focused ion beam scanning electron microscope (FIB-SEM, Zeiss Crossbeam 550). EBSD samples were prepared using argon-ion etching on an argon ion polishing machine (Gatan Ilion II 697). EBSD scans were performed with a step size of 0.55 μm. The morphologies and composition distribution of inclusions and Nb-bearing phases were analyzed using a field-emission SEM (Zeiss Ultra Plus) equipped with an energy dispersive spectrometer (EDS, Oxford X-MAX 50). At least 30 different SEM micrographs were collected, from which the size and number of inclusions and Nb-bearing phases were measured and counted. Further characterization of the inclusion@Z core-shell structure was performed on an aberration-corrected scanning transmission electron microscope (STEM, JEOL JEM-ARM200F) operated at 200 kV. The site-specific thin TEM foil was cut along the core-shell structure using an FIB (FEI Helios 600i) lift-out procedure. A polishing treatment was performed at 5 kV to further thin and clean the cross-section of the foil. Final polishing was repeated to ensure that the FIB lift-out was electron transparent. To detect the concentration distribution of elements around the Z phase, line profile analysis across the interface of the Z phase and matrix was performed using the STEM with the associated EDS. The nanohardness and elastic modulus of the matrix and Z phase were measured by a nanoindentation instrument (Bruker PI89) with a diamond-made triangular pyramidal indenter. The loading force was 1000 μN, and the loading time, holding time, and unloading time for each point were all 0.05 s. The final nanohardness and elastic modulus of each phase were given by the averages of at least five parallel indentations.

### Electrochemical and immersion corrosion tests

Electrochemical corrosion tests were conducted using a Gamry Reference 600 potentiostat with a standard three-electrode system. A platinum plate and a saturated calomel electrode (SCE) served as the counter and reference electrodes, respectively. The DSS samples were used as the working electrode, which were prepared according to the following method[64]. The samples were sealed with epoxy resin leaving an exposed area of 1.0 cm². The sample surface was ground with SiC papers to 2000 grit, cleaned with deionized water and ethanol, and finally dried in warm air. Electrochemical measurements were carried out in double-concentration simulated seawater at 72 °C (pH 8.2)[65], which can simulate the harshest corrosion conditions in the low-temperature multistage flash evaporation process of seawater

desalination equipment[47]. Prior to each measurement, the sample was polarized in the solution at −1.0 $V_{SCE}$ for 300 s. Then, the open circuit potential (OCP) was monitored for 1800 s until a steady state was reached. Subsequently, both potentiodynamic polarization and cyclic polarization tests were performed at a scan rate of 0.333 mV·s$^{-1}$ from −250 mV below the OCP to the anodic direction. When the current density reached 1 mA·cm$^{-2}$, the potentiodynamic polarization tests were terminated, and reverse scanning was started for the cyclic polarization tests. The pitting potential was defined as the potential at which the current density reached 100 μA·cm$^{-2}$, and the protective potential was recorded when the backward scan curve intersected with the forward scan curve. SEM observations were performed on the inclusions and inclusion@Z core-shell structure after potentiodynamic polarization tests were stopped at 100 mV, 200 mV, and pitting potential. DL-EPR tests were conducted in a 2 M $H_2SO_4$ + 0.75 M HCl + 0.01 M KSCN solution at 30 °C with a scan rate of 1 mV·s$^{-1}$. During the DL-EPR tests, the working electrodes were first kept at the OCP for 600 s. Subsequently, they were anodically polarized to +200 mV$_{SCE}$ and held for 120 s (forward scan) and then cathodically polarized to the OCP (reverse scan). The peak activation current density ($I_a$) and peak reactivation current density ($I_r$) were recorded during the forward and reverse scans, respectively. The degree of sensitization ($R$ value) was calculated by $R = (I_r/I_a) \times 100$. SEM observations were performed on the samples after DL-EPR tests to examine the degree of IGC. To ensure reproducibility and reliability, electrochemical tests were performed on at least eight samples of each steel.

Immersion corrosion tests were conducted following the ASTM G48 standard[66]. Several samples with dimensions of 50 mm × 25 mm × 5 mm were cut from the solution-treated DSS plates, and all their surfaces were ground with 1200 grit silicon carbide paper, followed by rinsing in deionized water, ultrasonic cleaning in ethanol, and drying in hot clean air. Subsequently, the prepared samples were immersed in 6% $FeCl_3$ solution at 50 °C. After immersion tests, the corrosion product on the sample surface was removed by a derusting solution (ISO 8407: 2009) through ultrasonic cleaning for 2 min, and then, the samples were rinsed in deionized water, ultrasonically cleaned in ethanol, and thoroughly dried. Moreover, the quantity, diameter, and depth of corrosion pits were observed and statistically counted by using a confocal laser scanning microscope (CLSM, Olympus OLS4100). Additionally, the weight of each sample before and after immersion was measured using an accurate electronic balance (accuracy of 0.1 mg), and the corrosion rate ($V_C$) was calculated by Eq. (7). To ensure reliability, three parallel samples were subjected to immersion corrosion to obtain the average corrosion rate of each steel.

$$V_C = \frac{87600 \times (W_0 - W_1)}{\rho A t} \tag{7}$$

where $W_0$ (g) and $W_1$ (g) represent the weights before and after the immersion test, respectively; $\rho$ (g·cm$^{-3}$) is the sample density; $A$ (cm$^2$) is the sum of the six surface areas of the sample; and $t$ (h) is the immersion time.

Ex situ SEM and atomic force microscopy (AFM, NT-MDT NEXT II) observations were performed on the inclusions and inclusion@Z core-shell structure before and after the immersion test. The samples for AFM observation were prepared in the same manner as those used for microstructure characterization mentioned above. AFM measurements were performed using a silicon probe coated with a thin PtIr layer in tapping mode to record the surface topography. In addition, scanning Kelvin probe force microscopy (SKPFM) was applied to measure the Volta potential difference among the inclusions, Z phase, and steel matrix. SKPFM measurements were taken in amplitude modulation mode using a scan rate of 1 Hz and a scan size of 6 μm. During the scanning process, the probe tip was lifted to -50 nm above the sample surface to measure the potential distribution. All AFM and SKPFM measurements were conducted in air at room temperature, and the related results were analyzed by Image Analysis software.

## Passive film analysis
To assess the influence of Nb on the passive film composition, X-ray photoelectron spectroscopy (XPS) measurements were performed using an ESCALAB250 (Thermo Fisher Scientific) equipped with an Al Kα X-ray source. The DSS samples used as working electrodes were gradually ground and polished with 1.5 μm diamond paste. To prepare the passive film, the working electrodes were first cathodically polarized at −1.0 $V_{SCE}$ for 300 s and then immediately potentiostatically polarized at −50 mV$_{SCE}$ for 2 h in double-concentration simulated seawater at 72 °C (pH 8.2). Subsequently, Ar$^+$ sputtering was performed on the sample surface to obtain the depth profile information of the passive film. The passive film spectra were analyzed by using XPSPEAK 4.0 software and an associated database (NIST X-ray Photoelectron Spectroscopy Database, https://srdata.nist.gov/xps/Default.aspx). The binding energies of all spectra were calibrated according to that of the C 1 $s$ peak (284.6 eV).

## Tensile test
Plate-type tensile samples (with a gauge length of 25 mm, a width of 4 mm, and a thickness of 2 mm) were machined from the solution-treated DSS plates along the rolling direction. Room-temperature tensile tests were conducted at a constant loading rate of 1 mm·min$^{-1}$ using a SHIMAZU AGS-X (loading capacity: 100 kN) electronic universal testing machine equipped with an extensometer. The strength and elongation of each steel were given by the averages of at least three parallel measurements.

## Data availability
The data that support the findings of this study are available from the corresponding author upon request. Source data are provided with this paper.

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

## Acknowledgements

This work was funded by the National Natural Science Foundation of China [Nos. 52325406 (H.L.), 52374334 (S.Z.), U1960203 (H.L.), 52004061 (S.Z.)], Science Fund for Distinguished Young Scholars of Liaoning Province [Grant No. 2023JH6/100500008 (H.L.)], Fundamental Research Funds for the Central Universities [No. N2225015 (S.Z.)], China Postdoctoral Science Foundation [No. 2021T140100 (S.Z.)], and Program of Introducing Talents of Discipline to Universities [Grant No. B21001 (Z.J.)]. We also acknowledge CITIC Metal Co., Ltd. and Shanxi Taigang Stainless Steel Co., Ltd. for the financial supports. Special thanks are due to the instrumental analysis from Analytical and Testing Centre, Northeastern University.

## Author contributions

S.Z. and H.L. conceived the idea and designed experiments. S.Z. and Y.L. prepared the materials and performed microstructure characterization, mechanical experiments, electrochemical and immersion corrosion tests. H.F. carried out the passive film analysis and SKPFM experiments. H.Z. conducted the theoretical calculations and corresponding analyses. W.Z. and G.L. prepared the industrial S32205 DSSs. S.Z., H.L., H.F., Z.J., T.Z., and Y.L. analysed the experimental data and co-wrote the manuscript. All authors contributed to the discussions and manuscript preparation.

## Competing interests

The authors declare no competing interests.
