## [Peer Review File · Nature Communications]

Design for improving corrosion resistance of duplex stainless steels by wrapping inclusions with niobium armourReviewers' comments:

Reviewer #1 (Remarks to the Author):

I would like to sign my reports to authors.

Reviewer #2 (Remarks to the Author):

The work presented concerns the development of a metallurgical approach for the resolution of localised corrosion of duplex stainless steels due to the presence of inclusionary impurities. The subject is very original, it is an alternative to deoxidation and desulphurisation treatments. The solution proposed here consists in embedding the inclusions to isolate them from the matrix. Thus, this treatment by adding additive elements avoids the galvanic coupling between the matrix. The authors of this very innovative solution propose a work that :

- Selects the alloying element that will enable the inclusionary impurities to be isolated (pre-study by thermocalc)
- Validates the coating phenomenon by microstructural analysis (SEM and statistical analysis)
- Proposes a corrosion study to estimate the corrosion resistance efficiency of the new alloy (polarisation curve, pitting potential analysis).

Thermocalc demonstrates that Nb is a good cladding element, which is shown by physicochemical analysis. The SEM images show good coverage of the inclusions. Unless the reviewer is mistaken, the paper seems to tell us that about 30% of the inclusions are not coated (Fig 2.e).

Several times in the paper it is stated that the Z-phase is corrosion resistant. It should rather be stated that the inclusionary phase corrodes before the Z-phase. There is no evidence that the Z-phase is stable because there is no polarisation curve backsweep that would indicate any repassivation. To be more precise. Let us remember that at the core of the material, obviously the inclusions do not corrode, however the inclusions which will be on the surface, still undergo corrosion and the little difference in the breakdown potentials of figure 5b and 5c, seems to remain in an uncertainty which is quite classical in electrochemistry.

Thus, to be fully convincing about the effect of this solution on corrosion, it would be necessary from the reporter's point of view :

- ensure a return scan of the polarisation curve to find out whether a passivation phenomenon is recorded,
- test the durability of the Z-phase. Indeed, corrosion is an energy release effect. If one phase corrodes, then the other phases are less exposed to dissolution. Once all the inclusionary phases have dissolved, it is not impossible that the Z-phase will not corrode.
- Finally, once the inclusion has dissolved, what effect does local acidification in the pit have on the Z-Phase?

In conclusion, in the opinion of the reviewer, the corrosion results are not complete and convincing enough to be published as such. Furthermore, it is not possible to conclude on the condition of the passive film by presenting only fig 4, the XPS analyses must be presented and discussed. It must be shown that once the dissolution of inclusionary impurities is achieved, the Z-phase will not corrode, and that discontinuities in the passive film will not be the source of corrosion sites. Therefore I propose to reject this paper. I suggest to the authors to present this original approach in a journal from the metallurgical point of view (without corrosion) and to consolidate the corrosion results to make them more convincing and to resubmit them to the journal.

Additional comments :

Line 69-70 not clear

Requirement 5 on page 4 is not sufficient as the ratio of galvanic corrosion surfaces must also be considered.

Reviewer #3 (Remarks to the Author):

The research reported in the manuscript looks at first sight very promising, with the title catching my initial attention, pointing towards an interesting approach to improve the corrosion resistance of duplex stainless steel. The approach overall is quite interesting, but the manuscript is lacking several key aspects to bring it up to the understanding required here. There are a number of aspects/questions that need to be addressed – see below:

(i) Most references listed seem to focus on materials not really relevant for the reported study; references to Nb modified steels/HSLA steel/other matrices is provided, but more importantly, citations dealing with Nb or Ti in duplex stainless steels are not discussed here. The effect of minor alloying elements (including Nb, Ti, Mo, W etc.) on duplex stainless steel has been studied quite extensively and must be discussed (for example: [1,2,3,4]). There are reports in literature of Nb-containing duplex stainless steel, which report a deleterious effect on the corrosion behaviour [1]. My experience on Nb-modified austenitic stainless steel, with the Nb forming NbC/NbCN in these alloys, showed typically a far higher localised corrosion susceptibility to pitting and crevice corrosion; which brings me to the second drawback of the proposed approach. The observation here is quite interesting, but more information about the mechanisms is required.

(ii) The chosen valve metals here for modifying duplex stainless steel are strong carbide and nitride formers, which is clearly evident in Fig.1. This results in the formation of nitrides at relatively high temperatures, due to the inherently high N-content present in duplex stainless steels. The Nb clearly forms (Cr, Nb)N in Fig.1(c), which then seems to convert completely into Z-phase which is surprising for me. I am still not clear what the driving force here is to convert (Cr,Nb)N into Z-phase. Did I miss anything here? The Z-phase is defined in the manuscript as (Nb, Cr, Mo)N-bearing precipitate? So what exactly do you define as Z-phase here vs. (Cr,Nb)N (Fig.1c vs. Fig. 6; where is your stable (Cr,Nb)N phase here?) vs. (Nb,Cr,Mo)N precipitate?

In Supp. Fig.6 the argument with Z phase being able to wrap around MnS phase does not hold here in this case, with MnS precipitating at lower temperatures compared to the apparent Z-phase here. The primary inclusions are present (Mg-Al-O), with the ferrite and austenite then forming at lower temperatures, followed by “Z-phase” formation.

Nb is a ferrite former, with also a secondary effect to bind to nitrogen, taking N away from austenite formers. N is also believed to be responsible for the corrosion resistance of the austenite phases. This will negatively affect the “equilibrium” phase ratio here, but again this is not discussed at all. The phase ratio is reported as 50:50, which might be related to non-equilibrium cooling conditions.

(iii) Corrosion - Fig.3 (a) indicates better pitting corrosion behaviour for pit growth (higher breakaway potential), but similar pit nucleation potentials for meta-stable pits. All materials seem to show meta stable pits forming at around +200 to +300 mV above the rest potential (E_{Corr}) in these test conditions, with stable pits then more likely growing in samples without Nb additions. But current spikes of Nb containing microstructures are clearly visible in Fig.3a at these low potentials. It looks like that pits are still initiated, but far fewer in number for Nb containing microstructures, reducing the probability to transition to stable pitting. This observation is interesting.

Figs. 3 & 5 clearly show that the cathodic and anodic branches of the electrochemical tests are nearly superimposed on each other, yielding the same passive current density and (most likely) very similar i_{corr} values (just be visual approximation). The increase in corrosion rate in Fig. 3c is then only accessible if localised corrosion is considered here (which is most likely the case for $FeCl_3$ exposure). Also, I would not expect for Nb to play a role in the passive film composition with such a small concentration and characteristic of atom.

I am really surprised that just one "standard composition" was used for the ThermoCalc simulations (Suppl.tab1 & tab.5) rather than the exact chemical compositions. For using the standard composition, I would expect to use the full range (min. – to max.) compositions of key elements.; some of the values used may significantly affect the overall precipitation behaviour.

The work reminded me on some of the earlier work on precipitation strengthened Ni-base alloys 718 where different (Al+Ti)/Nb -alloy ratios were used to create sandwich morphologies of the strengthening phases (γ'') and (γ'). This was back then an interesting approach, that was unfortunately not developed further (to my knowledge). [5]

[1] Filho, A.I., et al.: Effect of niobium in the phase transformation and corrosion resistance of one austenitic-ferritic stainless steel. *Mat.Res.* Vol 17, No.4.
<https://doi.org/10.1590/1516-1439.190113>

[2] Rossitti, S.M. & Rollo, J.M. Phase precipitation in a niobium-containing cast duplex stainless steel," *Metalurgia & Materials ABM* Vol. 54, 293-302, 1998.

[3] Gaurav, V. et al.: A Computational Approach to Examine Effect of Nb/Ti Doping on the Precipitation Behavior of Cast Super Duplex Stainless Steel, *Journal of Mat.Eng.&Performance*, Vol.17, 2023

[4] Gaurav, V. et al.: Influence of Nb Addition on Sliding Wear Behavior of 25 Cr 7 Ni Cast Austenitic-Ferritic Steel, *Journal of Mat.Eng.&Performance*, Vol. 31, p. 2043–2056, 2022.

[5] Cozar, R. & Pineau, A.: Morphology of γ' and γ'' Precipitates and Thermal Stability of Inconel 718 Type Alloys. *Metallurgical Transactions A*, Vol. 4, No. 1,p.47-59, 1973.

Point-by-point response to reviewers' comments (NCOMMS-22-44107)

We sincerely thank the reviewers for carefully reading our manuscript and for these constructive comments which significantly strengthen the presentation, significance and impact of our work. Detailed point-by-point response to the reviewers' comments and suggestions are summarized below. Accordingly, we have carefully revised the manuscript and supplementary materials, and all changes are marked in yellow.

Reviewer #1

Comments: Authors of this article have proposed a novel strategy of wrapping deleterious inclusions with corrosion-resistant niobium armour by doping a very small amount of niobium (0.25 wt%) in duplex stainless steels (DSSs). They have performed both theoretical and experimental work to explain the effectiveness of their strategy to largely increase the corrosion resistance of DSSs. I agree to the statement claimed in the abstract: Our strategy overcomes the long-standing problem of “corrosion failure caused by inclusions”, and it is verified as a universal technology in a series of DSSs and industrial production.

This work is of significance to the field of steels. Conventional wisdom often uses corrosion-resistance elements dissolved in the matrix lattice to increase corrosion resistance. In comparison, the present work uses niobium armour to warp deleterious inclusions to increase corrosion resistance.

The theoretical and experimental work performed in this article well supports the conclusions. To the best of my knowledge, I have found no flaws in the data analysis, interpretation and conclusions.

The methodology is sound. The work meets the expected standards in my field.

The detail provided in the methods is enough for the work to be reproduced.

Based on these judgements, I suggest that this article be accepted for publication in Nature Communications.

Response: We thank the reviewer a lot for recognizing the broad interest and high impact of our work, as well as the high affirmation of our strategy and the strength of our novel conclusions. For a long time, “corrosion failure caused by inclusions” has become a bottleneck problem in the scientific and engineering practices of steel corrosion protection. Our work overcomes this thorny problem, which is of significance to the field of steels.

Reviewer #2

Comment 1: The work presented concerns the development of a metallurgical approach for the resolution of localised corrosion of duplex stainless steels due to the presence of inclusionary impurities.

The subject is very original, it is an alternative to deoxidation and desulphurisation treatments. The solution proposed here consists in embedding the inclusions to isolate them from the matrix. Thus, this treatment by adding additive elements avoids the galvanic coupling between the matrix.

The authors of this very innovative solution propose a work that:

- Selects the alloying element that will enable the inclusionary impurities to be isolated (pre-study by thermocalc)
- Validates the coating phenomenon by microstructural analysis (SEM and statistical analysis)
- Proposes a corrosion study to estimate the corrosion resistance efficiency of the new alloy (polarisation curve, pitting potential analysis).

Response: We thank the reviewer a lot for high recognition of the innovation and impact of our work. Over the past decades, extensive efforts have been paid to address the localised corrosion of duplex stainless steels (DSSs) due to the presence of inclusionary impurities, but none of them was very effective. In this work, we innovatively proposed a strategy of wrapping deleterious inclusions with corrosion-resistant niobium armour (Z phase), which can be an ideal alternative to deoxidation and desulphurisation treatments, significantly improving the corrosion resistance of DSSs.

Comment 2: Thermocalc demonstrates that Nb is a good cladding element, which is shown by physicochemical analysis. The SEM images show good coverage of the inclusions.

Unless the reviewer is mistaken, the paper seems to tell us that about 30% of the inclusions are not coated (Fig 2.e).

Response: We greatly appreciate the reviewer for the valuable comments. As shown in Fig. 2e, the proportion of inclusions wrapped by the Z phase in 0.25Nb steel is indeed 75.8% (i.e., 24.2% of the inclusions are not coated). Theoretical calculation results (Fig. 1c and Supplementary Table 2) demonstrate that Nb microalloying shows considerable promise in terms of wrapping inclusions with Nb-bearing phase, which is mainly achieved through heterogeneous nucleation. However, during actual preparation process of the steel, heterogeneous nucleation is affected by many factors, such as solidification cooling conditions, hot working and heat treatment

techniques. Thus, for a few inclusions, there may be no heterogeneous nucleation of Nb-bearing phase around them.

Inspired by the reviewer's comment, we explored the effect of inclusions wrapped by the Z phase on pit growth using the Gumbel extreme value distribution^[r1]. The depths of 15 deepest pits were arranged by small to large order, and the probability of pit depths $F(Y)$ can be calculated by^[r2]:

$$F(Y) = 1 - \frac{n}{N+1}$$

where n is the rank in the ordered pit depth, and N is the total number of selected pits. The reduced variant (Y) can be calculated by^[r2]:

$$Y = -\ln\{-\ln[F(Y)]\}$$

The maximum pit depth can be calculated by the Gumbel type extreme value distribution expressed in the following form^[r2]:

$$\begin{aligned} Pit_{\max} &= \mu + \alpha \ln T \\ T &= S/s \end{aligned}$$

where Pit_{\max} is the maximum pit depth, μ is the central parameter (the most frequent value), and α is the scale parameter that defines the width of the distribution. S is the area over which a prediction is to be made, s is the statistical area of the sample, and T is the ratio of predicted area to statistical area.

Supplementary Fig. 6a and **b** show the cumulative probability of pit depths and the Gumbel probability plots of S32205 DSSs after immersion corrosion in 6% FeCl₃ solution at 50 °C for 12 h. The proportion of inclusions wrapped by the Z phase in 0.10Nb and 0.25Nb steels are 49.3% and 75.8% (**Fig. 2e**), indicating that the effective areas (s) used for statistics in 0.10Nb and 0.25Nb steels are 50.7% and 24.2% of that in 0Nb steel, respectively. Therefore, the Gumbel distribution parameters ($\mu = 131.6$ and $\alpha = 46.5$) and the maximum pit depth (243.5 μm) of 0Nb steel, together with the area proportions of 0.10Nb and 0.25Nb steels are used to predict the maximum pit depth of 0.10Nb and 0.25Nb steels, as follows:

$$\begin{aligned} Pit_{\max}^{0\text{Nb}} &= 131.6 + 46.5 \ln T \\ Pit_{\max}^{0.10\text{Nb}} &= 131.6 + 46.5 \ln (0.507T) \\ Pit_{\max}^{0.25\text{Nb}} &= 131.6 + 46.5 \ln (0.242T) \end{aligned}$$

Supplementary Fig. 6c illustrates that the predicted values for the maximum pit depth of 0.10Nb and 0.25Nb steels are 211.9 and 177.5 μm , respectively, which are close to the measured values (192.5 and 164.0 μm). The errors for 0.10Nb and 0.25Nb steels are 9.1% and 7.6%, respectively. In summary, the pit growth of three S32205 DSSs can be well modelled by the

Gumbel extreme value distribution. Once the inclusions are wrapped by the Z phase, the proportion of inclusions that may induce deep pits is considerably decreased, therefore, the maximum pit depth of Nb-bearing steels becomes significantly smaller.

Supplementary Fig. 6 Gumbel extreme value distribution. **a** Cumulative probability of pit depths and **b** the Gumbel probability plots of S32205 DSSs after immersion corrosion in 6% FeCl₃ solution at 50 °C for 12 h. **c** Comparison between the predicted and measured values for the maximum pit depth of 0.10Nb and 0.25Nb steels.

[r1] Valor, A., Caleyó, F., Alfonso, L., Rivas, D. & Hallen, J. M. Stochastic modeling of pitting corrosion: A new model for initiation and growth of multiple corrosion pits, *Corros. Sci.* **49**, 559–579 (2007).

[r2] Zhang, T. et al. Corrosion of pure magnesium under thin electrolyte layers, *Electrochim. Acta* **53**, 7921–7931 (2008).

Revision in the manuscript:

Meanwhile, the Gumbel extreme value distribution is used to explore the effect of Nb microalloying on pit growth (**Supplementary Note 1** and **Supplementary Fig. 6**). The results show that the pit growth of three S32205 DSSs can be well modelled by the Gumbel extreme value distribution. Once the inclusions are wrapped by Z phase, the proportion of inclusions that may induce deep pits is considerably decreased, therefore, the maximum pit depth of Nb-bearing steels becomes significantly smaller. (**Page 10, Lines 193-199**)

Comment 3: Several times in the paper it is stated that the Z-phase is corrosion resistant. It should rather be stated that the inclusionary phase corrodes before the Z-phase. There is no evidence that the Z-phase is stable because there is no polarisation curve backsweep that would indicate any repassivation.

To be more precise. Let us remember that at the core of the material, obviously the inclusions do not corrode, however the inclusions which will be on the surface, still undergo corrosion and the little difference in the breakdown potentials of figure 5b and 5c, seems to remain in an uncertainty which is quite classical in electrochemistry.

Thus, to be fully convincing about the effect of this solution on corrosion, it would be necessary from the reporter's point of view:

- ensure a return scan of the polarisation curve to find out whether a passivation phenomenon is recorded.
- test the durability of the Z-phase. Indeed, corrosion is an energy release effect. If one phase corrodes, then the other phases are less exposed to dissolution. Once all the inclusionary phases have dissolved, it is not impossible that the Z-phase will not corrode.
- Finally, once the inclusion has dissolved, what effect does local acidification in the pit have on the Z-Phase?

Response: Thanks for the reviewer's very insightful suggestions. It is indeed necessary to directly compare the corrosion resistance of Z phase, inclusions, and the steel matrix, so as to confirm the durability of Z phase. Thus, we further observed the corrosion morphologies of inclusions and inclusion@Z core-shell structure after potentiodynamic polarization tests stopped at 100 mV, 200 mV, and pitting potential (Fig. 3b and c). At 100 mV and 200 mV, visible metastable pits formed around the individual inclusions in 0Nb steel and further expanded into large-size corrosion pits at the pitting potential. For the inclusion@Z core-shell structure in 0.25Nb steel, although metastable pits also formed around the core inclusion at low potential, they did not expand significantly at the pitting potential. These phenomena indicate that the Z phase is indeed very corrosion resistant, so that local acidification in the pit caused by inclusion dissolution will not induce apparent corrosion of Z phase. Meanwhile, the corrosion-resistant Z phase shell effectively prevents the development of metastable pit into stable pit.

Fig. 3 Electrochemical and immersion corrosion behaviour of S32205 DSSs. Electrochemical corrosion behaviour in double-concentration simulated seawater at 72 °C (pH 8.2): **a** potentiodynamic polarization curves and corrosion morphologies of **b** 0Nb and **c** 0.25Nb steels.

To further explore the durability of Z phase, we carried out long-term immersion corrosion tests in 6% FeCl₃ solution at 50 °C (Fig. 4d and Supplementary Fig. 9). After immersion for 10 d, the core MnS part in the MgAl₂O₄-MnS@Z core-shell structure dissolved, and several pieces of steel matrix were also corroded. However, no visible corrosion sign was found on or around Z phase. These facts indicate that the corrosion resistance of Z phase is much higher than those of the steel matrix and inclusions, and local acidification in the pit will not induce corrosion of Z phase. Based on the above electrochemical and long-term immersion corrosion results, we can firmly confirm that Z phase is indeed a highly corrosion resistant precipitate that can effectively prevent corrosion caused by inclusions without inducing other types of corrosion.

Fig. 4 Characterization of S32205 DSSs before and after immersion corrosion. SEM morphologies (left) and AFM topographies (right) before (**a,c**) and after (**b,d**) immersion corrosion in 6% FeCl₃ solution at 50 °C for **b** 12 h and **d** 10 d. **a,b** MgAl₂O₄-MnS inclusion in 0Nb steel and **c,d** MgAl₂O₄-MnS@Z core-shell structure in 0.25Nb steel.

Supplementary Fig. 9 Characterization of 0.25Nb S32205 DSS after immersion corrosion. SEM morphologies of inclusion@Z core-shell structure in the steel after immersion corrosion in 6% FeCl₃ solution at 50 °C for 10 d.

Regarding the reviewers' concern about the results in Fig. 5b and c, we have provided the cumulative probability distribution of pitting potentials (Supplementary Fig. 5b and c), which firmly indicates that Nb microalloying is also an applicable strategy for improving the corrosion resistance of S32101 and S32750 DSSs by wrapping deleterious inclusions with niobium armour (Z phase).

Supplementary Fig. 5 Cumulative probability distribution of pitting potential. Cumulative probability distribution of pitting potential of various Nb-free and Nb-bearing DSSs in double-concentration simulated seawater at 72 °C (pH 8.2): **b** S32101 and **c** S32750.

In addition, we have carefully considered the suggestion of conducting cyclic polarisation with backward polarization. As mentioned above, metastable pits mainly initiated around the core inclusion of inclusion@Z core-shell structure in Nb-bearing steels. These pits could not re-passivate during backward scan because the Z phase surrounding them was always stable. Even so, these metastable pits did not expand into large-size corrosion pits due to the wrap by highly corrosion-resistant Z phase.

Revision in the manuscript:

Fig. 3b and c show the corrosion morphologies of inclusions and inclusion@Z core-shell structure after potentiodynamic polarization tests stopped at 100 mV, 200 mV, and pitting potential. At 100 mV and 200 mV, visible metastable pits formed around the individual inclusions in 0Nb steel and further expanded into large-size corrosion pits at the pitting potential. For the inclusion@Z core-shell structure in 0.25Nb steel, although metastable pits also formed around the core inclusion at low potential, they did not expand significantly at the pitting potential. These phenomena indicate that the Z phase is indeed very corrosion resistant, so that local acidification in the pit caused by inclusion dissolution will not induce apparent corrosion of Z phase. Meanwhile, the corrosion-resistant Z phase shell effectively prevents the development of metastable pit into stable pit. **(Pages 9-10, Lines 180-190)**

After immersion for 10 d, the core MnS part of the MgAl₂O₄-MnS@Z core-shell structure in 0.25Nb steel dissolved, and several pieces of steel matrix were also corroded. However, no visible corrosion sign was found on or around the Z phase (Fig. 4c, d and **Supplementary Fig. 9**). These facts indicate that the corrosion resistance of Z phase is much higher than those of the steel matrix and inclusions, and local acidification in the pit will not induce corrosion of Z phase. This is because the (Nb, Cr, Mo, N)-rich Z phase is a corrosion-resistant precipitate with a Volta potential (i.e., electrochemical activity¹¹) similar to that of the steel matrix (Fig. 4f and g), and no micro-crevice forms around Z phase. **(Page 11, Lines 219-226)**

Comment 4: In conclusion, in the opinion of the reviewer, the corrosion results are not complete and convincing enough to be published as such. Furthermore, it is not possible to conclude on the condition of the passive film by presenting only fig 4, the XPS analyses must be presented and discussed. It must be shown that once the dissolution of inclusionary impurities is achieved, the Z-phase will not corrode, and that discontinuities in the passive film will not be the source of corrosion sites.

Response: We thank the reviewer for the very pertinent comment. As replied above, we have carried out further electrochemical and long-term immersion corrosion tests to make the corrosion results more complete and convincing. The results (Figs. 3c and 4d) definitely show that even the inclusion is dissolved, the Z phase will not corrode. Regarding the condition of passive film, we have provided XPS analysis results (Supplementary Fig. 7). It is apparent that there are no significant differences in the O and Cr concentrations or Cr-bearing components between the passive films of Nb-free and Nb-bearing steels. Additionally, there is also no apparent enrichment of Nb-bearing oxides in the passive film of Nb-bearing steels. Thus, it can be concluded that the addition of Nb has little influence on the composition of passive film. (Pages 10-11, Lines 203-208)

Regarding the corrosion sites on the Nb-bearing steel, the core MnS part in the MgAl₂O₄-MnS@Z core-shell structure in 0.25Nb steel preferentially dissolved, due to the lower Volta potential and discontinuities in passive film. However, the (Nb, Cr, Mo, N)-rich Z phase is a corrosion-resistant precipitate with a Volta potential similar to that of steel matrix (Fig. 4f and g), and no micro-crevice forms around Z phase. Thus, the Z phase shell would not corrode even after long-term immersion corrosion tests. Under such a condition, discontinuities in the passive film above the core inclusions will not be the source of corrosion sites.

Supplementary Fig. 7 Characterization of passive films. Compositions of passive films formed on S32205 DSSs: **a** O concentration, **b** Cr concentration, **c** XPS spectra of Cr, and **d** XPS spectra of Nb.

Comment 5: Therefore, I propose to reject this paper. I suggest to the authors to present this original approach in a journal from the metallurgical point of view (without corrosion) and to consolidate the corrosion results to make them more convincing and to resubmit them to the journal.

Response: Again, we highly appreciate the reviewer for all above valuable comments and constructive suggestions which are important for improving the integrity and impact of our work. Accordingly, we have strengthened the corrosion results and fully confirmed the credibility of our strategy, as replied above. We have revised the manuscript in full accordance with the reviewer's comments and suggestions. We hope the revision is satisfactory and is now suitable for publication in Nature Communications.

Comment 6: Additional comments:

Line 69-70 not clear

Requirement 5 on page 4 is not sufficient as the ratio of galvanic corrosion surfaces must also be considered.

Response: Thanks for the reviewer's valuable suggestions. The descriptions in original Line 69-70 have been revised as follows. "(3) the precipitate should exist stably instead of dissolving into the steel matrix during hot working and heat treatment procedures. To achieve the second essential condition, the other three requirements need to be met:". **(Page 5, Lines 85-88)**

We agree with the reviewer's suggestion that the ratio of galvanic corrosion surfaces must be considered in requirement (5). Indeed, both the potential difference between the galvanic couple and the area ratio of anode and cathode are key factors affecting galvanic corrosion. To avoid the occurrence of galvanic corrosion, it is necessary to prevent forming a galvanic couple composed of a small anode and a large cathode. Accordingly, we have supplemented this consideration to the revised manuscript.

Revision in the manuscript:

(5) the potential difference between the precipitate and surrounding matrix should be small, especially the precipitate's potential should not be lower than the matrix to prevent forming a galvanic couple composed of a small anode and a large cathode, thus avoiding galvanic corrosion¹¹. **(Page 5, Lines 89-92)**

In the present work, the electrode potential of the Z phase is basically the same as that of the surrounding matrix (Fig. 4f and g), so they will not form a corrosive galvanic couple composed of a small anode and a large cathode. Namely, no galvanic corrosion occurs between them. **(Pages 15-16, Lines 316-319)**

Reviewer #3

Comment 1: The research reported in the manuscript looks at first sight very promising, with the title catching my initial attention, pointing towards an interesting approach to improve the corrosion resistance of duplex stainless steel. The approach overall is quite interesting, but the manuscript is lacking several key aspects to bring it up to the understanding required here. There are a number of aspects/questions that need to be addressed – see below:

Response: We thank the reviewer a lot for recognizing the broad interest and high prospect of our work. We also highly appreciate the reviewer for all the valuable comments and constructive suggestions which are important for improving the integrity and impact of our work. Accordingly, we have supplemented additional experiments, calculations, related descriptions and discussions to address all the reviewer's valuable comments and suggestions. Detailed point-by-point responses are summarized below.

Comment 2: (i) Most references listed seem to focus on materials not really relevant for the reported study; references to Nb modified steels/HSLA steel/other matrices is provided, but more importantly, citations dealing with Nb or Ti in duplex stainless steels are not discussed here. The effect of minor alloying elements (including Nb, Ti, Mo, W etc.) on duplex stainless steel has been studied quite extensively and must be discussed (for example: [1,2,3,4]). There are reports in literature of Nb-containing duplex stainless steel, which report a deleterious effect on the corrosion behaviour [1]. My experience on Nb-modified austenitic stainless steel, with the Nb forming NbC/NbCN in these alloys, showed typically a far higher localised corrosion susceptibility to pitting and crevice corrosion; which brings me to the second drawback of the proposed approach. The observation here is quite interesting, but more information about the mechanisms is required.

[1] Filho, A.I., et al.: Effect of niobium in the phase transformation and corrosion resistance of one austenitic-ferritic stainless steel. *Mat.Res.* Vol 17, No.4. <https://doi.org/10.1590/1516-1439.190113>

[2] Rossitti, S.M. & Rollo, J.M. Phase precipitation in a niobium-containing cast duplex stainless steel," *Metalurgia & Materials ABM* Vol. 54, 293-302, 1998.

[3] Gaurav, V. et al.: A Computational Approach to Examine Effect of Nb/Ti Doping on the Precipitation Behavior of Cast Super Duplex Stainless Steel, *Journal of Mat.Eng.&Performance*, Vol.17, 2023

[4] Gaurav, V. et al.: Influence of Nb Addition on Sliding Wear Behavior of 25 Cr 7 Ni Cast Austenitic-Ferritic Steel, *Journal of Mat.Eng.&Performance*, Vol. 31, p. 2043–2056, 2022.

Response: We highly appreciate the reviewer for his/her broad knowledges about the effect of microalloying elements on duplex stainless steel and also for the very pertinent suggestions.

We have further consulted extensive references on the application of microalloying elements in duplex stainless steels, including those recommended by the reviewer.

Regarding Nb microalloying in DSSs, the existing researches mainly includes the following aspects. (1) Nb can improve hardness, sliding wear resistance, cavitation erosion resistance, and slurry erosion resistance of DSSs through solid solution strengthening, NbC precipitation strengthening, and TWIP effect [r1-r5]. (2) Nb can retard the formation of chromium carbides and suppress the chromium depletion, thereby enhancing intergranular corrosion resistance of DSSs [r6, r7]. (3) Nb can inhibit the precipitation of detrimental phases (Σ , Cr_2N , and M_{23}C_6) in DSSs aged at 600 °C [r8]. However, Nb can promote the formation of Σ and Laves phases in DSSs aged at 850 °C, thus increasing the hardness and wear resistance but reducing the corrosion resistance [r9-r12].

Regarding Ti microalloying in DSSs, the existing researches mainly includes the following aspects. (1) Ti can inhibit the precipitation of chromium carbide, thereby improving the hot workability of DSS [r13]. (2) Ti can promote the formation of titanium silicides, increase ferrite phase fraction and decrease equiaxed grain size, thus increasing the strength of DSSs [r14, r15]. However, Ti can significantly enhance the embrittlement of DSS aged at 475 °C [r16]. (3) Ti can promote chromium enrichment in passive film, thereby improving corrosion resistance [r17]. But no passive film formed on (Ti,Cr)N, which induced crevice or crack around (Ti,Cr)N and further promoted pitting corrosion of DSS [r18].

In addition, there are some studies on improving the corrosion resistance of DSSs by adding other alloying elements such as Mo or W [r19-r22]. But these methods are not competent to solve the localised corrosion problem induced by inclusions.

In summary, although Nb and Ti microalloying have certainly been applied in DSSs, the relevant researches mainly focused on their effects on the precipitation of Nb/Ti-bearing phases, chromium carbides and intermetallic phases, and the corresponding hot workability, mechanical properties and corrosion resistance. To the best of our knowledge, there is no report on wrapping inclusion with corrosion-resistant precipitate through applying microalloying technology to improve the corrosion resistance of DSSs.

Regarding the report about Nb deteriorating the corrosion resistance of DSS (reference [1] recommended by the reviewer), this mainly occurred after the steel was aged at 850 °C. This aging treatment promoted the precipitation of Σ phase, and Nb addition promoted the formation of Laves phase. Σ phase in association with Laves phase, led to a decrease in the corrosion resistance of the steel. Generally, before application, DSSs are subjected to solid solution treatment to fully dissolve detrimental intermetallic phases (Σ and Laves phases,

etc.) back into the steel matrix, in order to avoid their adverse effects on mechanical properties and corrosion resistance. Therefore, after appropriate solid solution treatment, the above adverse effect of Nb on the corrosion resistance can be eliminated.

As mentioned by the reviewer, NbC or NbCN can induce localised corrosion of Nb-modified austenitic stainless steels [r23, r24]. The relevant reasons mainly include: the micro-galvanic coupling between NbC and austenite matrix, the micro-crevice at the interface between Nb-bearing phase and austenite matrix formed during deformation process, and the reduction of passive film stability due to N depletion around Nb(C,N). However, Niederhofer et al. [r25] reported that Nb(C,N) will not induce significant localised corrosion of Nb-modified austenitic stainless steel, because the steel contains a certain amount of Mo (~2.5%), which improves the stability of passive film.

In the present work, the electrode potential of Z phase is basically the same as that of the surrounding matrix (Fig. 4f and g), so no micro-galvanic corrosion occurs between them. Meanwhile, no micro-crevice forms at the Z phase/matrix interface (Fig. 2f and g), which is attributed to their basically similar nano-hardness and elastic modulus (**Supplementary Table 8**). Furthermore, the experimental steel contains a certain amount of Mo (~3.1%) and N (~0.16%), both of which can synergistically improve the passive film stability. Thus, the precipitation of Z phase should not significantly deteriorate the passive film stability. The XPS results (**Supplementary Fig. 7**) also indicate that Nb addition has no apparent effect on the passive film. Therefore, the Z phase is indeed highly corrosion resistant, and it will not induce localised corrosion of the steel, which has been fully confirmed by the results of electrochemical and immersion corrosion tests (Fig. 3, Fig. 4 and **Supplementary Fig. 9**). And the relevant mechanisms have been given in Discussion part. (**Pages 15-16, Lines 305-333**)

Revision in the manuscript:

However, although Nb and Ti microalloying have certainly been applied in DSSs, the relevant researches mainly focused on their effects on the precipitation of Nb/Ti-bearing phases, chromium carbides and intermetallic phases, and the corresponding hot workability, mechanical properties and corrosion resistance²³⁻³². In addition, some efforts were also tried to improve the corrosion resistance of DSSs by adding alloying elements such as Mo or W^{33,34}. But these methods are not competent to solve the localized corrosion problem induced by inclusions. To the best of our knowledge, there is no report on wrapping inclusion with corrosion-resistant precipitate through applying microalloying technology to improve the corrosion resistance of DSSs. (**Page 4, Lines 57-65**)

- [r1] Gaurav, V., Kumareshbabu, S. P. & Sankaranarayanan, S. R. Influence of Nb addition on sliding wear behavior of 25Cr7Ni cast austenitic-ferritic steel. *J. Mater. Eng. Perform.* **31**, 2043–2056 (2022).
- [r2] Bao, Y. F. et al. Cavitation erosion behavior of Nb strengthened duplex stainless steel surfacing layer. *J. Mater. Eng. Perform.* **31**, 10367–10377 (2022).
- [r3] Rajkumar, M., Babu, S. P. K., Vallimanan, A. & Mahendran, R. Room-temperature erosion behaviour of Nb-stabilized 27Cr-7Ni-Mo-W-N cast hyper-duplex stainless steel (Nb + CD3MWN - 7A). *J. Inst. Eng. India Ser. D* **100**, 83–90 (2019).
- [r4] Bao, Y. F. et al. Strengthening effect of Nb on microstructure and cavitation erosion behavior of duplex stainless steel surfacing layer. *J. Mater. Eng. Perform.* (2023). <https://doi.org/10.1007/s11665-023-07982-7>
- [r5] Gaurav, V., Sankaranarayanan, S. R., Kumareshbabu, S. P. & Vallimanan, A. Slurry erosion behavior of cast 25Cr austenitic-ferritic steel with niobium addition for subsea pipelines. *J. Mater. Eng. Perform.* **30**, 6224–6234 (2021).
- [r6] Rajkumar, M., Babu, S.P. K. & Nagaraj T. A. Intergranular corrosion characteristics of niobium stabilized 27Cr-7NiMo-W-N cast hyper duplex stainless steel. *Mater. Today Proc.* **27**, 2551–2555 (2020).
- [r7] Kumar, P. A., Babu, S. P. K. & Thirumaran, B. Studies on intergranular corrosion characteristics of cast duplex stainless steel stabilized with niobium. *J. Test. Eval.* **47**, 3690–3704 (2019).
- [r8] Gaurav, V., Kumareshbabu, S. P. & Sankaranarayanan, S. R. A computational approach to examine effect of Nb/Ti doping on the precipitation behavior of cast super duplex stainless steel. *J. Mater. Eng. Perform.* (2023). <https://doi.org/10.1007/s11665-023-07833-5>
- [r9] Filho, A. I., Cardoso, W. D., Gontijo, L. C., Silva, R. V. & Casteletti, L. C. Austenitic-ferritic stainless steel containing niobium. *Rem: Rev. Esc. Minas* **66**, 467–471 (2013).
- [r10] Cardoso, W. D. & Baptista, R. C. Laves phase precipitation and sigma phase transformation in a duplex stainless steel microalloyed with niobium. *Matéria Rio. J.* **27**, (2022).
- [r11] Filho, A. I., Silva, R. V., Cardoso, W. D. & Casteletti, L. C. Effect of niobium in the phase transformation and corrosion resistance of one austenitic-ferritic stainless steel. *Mater. Res.* **17**, 801–806 (2014).
- [r12] Rossitti, S. M. & Rollo, J. M. D. A. Phase precipitation in a niobium-containing cast duplex stainless steel. *Metal. Mater. ABM* **54**, 293–302 (1998).
- [r13] Merrick, H. F., Hayden, H. W. & Gibson, R. C. The effect of carbon and titanium on the hot workability of 25Cr-6Ni stainless steels. *Metall. Trans.* **4**, 827–832 (1973).
- [r14] Koztowski, R. H. Composite of austenitic-ferritic stainless steel. *J. Mater. Process. Technol.* **53**, 239–246 (1995).
- [r15] Wang, Q. M., Cheng, G. G. & Hou, Y. Y. Effect of titanium addition on as-cast structure and high-temperature tensile property of 20Cr-8Ni stainless steel for heavy castings. *Metals* **10**, 529 (2020).
- [r16] Yang, Z. Y., Su, J. & Wang, Y. M. Investigation on metallurgical factors controlling Charpy impact toughness in 1Cr21Ni5Ti duplex stainless steel. *J. Iron Steel Res. Int.* **16**, 73–79 (2009).
- [r17] Zhang, J. T., Hu, X. J. & Chou, K. Effects of Ti addition on microstructure and the associated corrosion behavior of a 22Cr-5Ni duplex stainless steel. *Mater. Corros.* **72**, 1201–1214 (2021).
- [r18] Hou, Y. Y., Nakamori, Y., Kadoi, K., Inoue, H. & Baba, H. Initiation mechanism of pitting corrosion in weld heat affected zone of duplex stainless steel. *Corros. Sci.* **201**, 110278 (2022).
- [r19] Tian, H. C., Cheng, X. Q., Wang, Y., Dong, C. F. & Li, X. G. Effect of Mo on interaction between α/γ phases of duplex stainless steel. *Electrochim. Acta* **267**, 255–268 (2018).
- [r20] Mesquita, T. J. et al. Lean duplex stainless steels-The role of molybdenum in pitting corrosion of concrete reinforcement studied with industrial and laboratory castings. *Mater. Chem. Phys.* **132**, 967–972

(2012).

[r21] Torres, C., Johnsen, R. & Iannuzzi, M. Crevice corrosion of solution annealed 25Cr duplex stainless steels: Effect of W on critical temperatures. *Corros. Sci.* **178**, 109053 (2021).

[r22] Park, C. J. & Kwon, H. S. Effects of aging at 475 °C on corrosion properties of tungsten-containing duplex stainless steels. *Corros. Sci.* **44**, 2817–2830 (2002).

[r23] Bai, G. S., Lu, S. P., Li, D. Z. & Li, Y. Y. Influences of niobium and solution treatment temperature on pitting corrosion behaviour of stabilised austenitic stainless steels. *Corros. Sci.* **108**, 111–124 (2016).

[r24] Clark, R. N., Searle, J., Martin, T. L., Walters, W. S. & Williams, G. The role of niobium carbides in the localised corrosion initiation of 20Cr-25Ni-Nb advanced gas-cooled reactor fuel cladding. *Corros. Sci.* **165**, 108365 (2020).

[r25] Niederhofer, P., Huth, S. & Theisen, W. The impact of cold work and hard phases on cavitation and corrosion resistance of high interstitial austenitic FeCrMnMoCN stainless steels. *Wear* **376-377**, 1009–1020 (2017).

Comment 3: (ii) The chosen valve metals here for modifying duplex stainless steel are strong carbide and nitride formers, which is clearly evident in Fig.1. This results in the formation of nitrides at relatively high temperatures, due to the inherently high N-content present in duplex stainless steels. The Nb clearly forms (Cr,Nb)N in Fig.1(c), which then seems to convert completely into Z-phase which is surprising for me. I am still not clear what the driving force here is to convert (Cr,Nb)N into Z-phase. Did I miss anything here? The Z-phase is defined in the manuscript as (Nb, Cr, Mo)N-bearing precipitate? So what exactly do you define as Z-phase here vs. (Cr,Nb)N (Fig.1c vs. Fig. 6; where is your stable (Cr,Nb)N phase here?) vs. (Nb, Cr, Mo)N precipitate?

Response: Thanks for the reviewer's very insightful and valuable comments. Firstly, let's explain the definition principles of Z phase and (Cr,Nb)N phase in this work. Z phase has a tetragonal structure with the lattice parameters of $a = 0.3037$ nm and $c = 0.7391$ nm. In this work, Z phase is defined as a (Nb, Cr, Mo)N-bearing precipitate according to its calculated composition (Fig.1f, 1050 °C: 51.63%Nb, 26.61%Cr, 6.76%Mo, and 8.60%N). In contrast, (Cr,Nb)N phase actually belongs to a NbN-type precipitate, which has a FCC structure with the lattice parameter of $a = 0.439$ nm. The definition of (Cr,Nb)N phase is also based on its calculated composition (Fig.1f, 1300 °C: 71.18%Nb, 14.38%Cr, and 12.58%N).

We are very sorry for not considering the transformation from (Cr,Nb)N to Z phase in our original submission, which caused confusion about this subject. To be honest, although there is indeed a transition from (Cr,Nb)N to Z phase in the **equilibrium calculation result (Fig. 1c)**, only Z phase precipitates in the **nonequilibrium calculation result (original Supplementary Fig. 6)**. Moreover, we did only observe Z phase in the sample after hot deformation-solution

treatment. These results make us mistakenly believe that Z phase directly formed in the actual preparation process, and there is no transformation from (Cr,Nb)N to Z phase. Under the important reminder of the reviewer, we have carefully reconsidered this issue and supplied additional calculations and experiments to clarify this confusion.

Firstly, we carefully verified the rationality of nonequilibrium calculation result. Indeed, there was always no (Cr,Nb)N precipitation in the result calculated by **Thermo-Calc software**. This may be due to the incomplete database of nonequilibrium in Thermo-Calc software. Thus, we recalculated the nonequilibrium solidification process of 0.25Nb steel using **FactSage software (Supplementary Fig. 10)**. We surprisingly found that (Cr,Nb)N instead of Z phase preferentially formed during solidification process. To further verify the reliability of the calculation result, we conducted SEM and TEM observations on the as-cast structure. It was found that the Nb-bearing phase formed in as-cast structure was mainly (Cr,Nb)N, thereby forming MgAl₂O₄@(Cr,Nb)N, MnS@(Cr,Nb)N and MgAl₂O₄-MnS@(Cr,Nb)N core-shell structures (**Supplementary Fig. 11**). (Here, we would like to apologize for a writing error: in original submission, the nonequilibrium solidification process of 0.25Nb S32205 DSS was actually calculated using Thermo-Calc software, but we mistakenly wrote it as FactSage software.)

Supplementary Fig. 10 Nonequilibrium solidification process of 0.25Nb S32205 DSS calculated by the Gulliver-Scheil model in FactSage software.

Supplementary Fig. 11 Characterization of inclusion@(Cr,Nb)N core-shell structures. Morphologies and SEM-EDS compositions (wt.%) of inclusion@(Cr,Nb)N core-shell structures in as-cast 0.25Nb S32205 DSS: **a** MgAl₂O₄@(Cr,Nb)N, **b** MnS@(Cr,Nb)N, and **c** MgAl₂O₄-MnS@(Cr,Nb)N core-shell structures. **d** TEM bright-field image and corresponding selected area electron diffraction pattern of (Cr,Nb)N.

Based on the calculation and experimental results, we can firmly believe that (Cr,Nb)N is the preferred Nb-bearing phase formed during solidification process. So, how did this phase convert into Z phase? For this issue, we noticed that the cast ingot experienced a heating process (1180 °C × 1 h) before hot working. Thus, we conducted heat treatment experiments at 1180 °C for different times to observe the transformation of Nb-bearing phase. From the phase composition analysis results (**Supplementary Fig. 12**), it can be determined that after heat treatment at 1180 °C for 60 min, the (Cr,Nb)N phase completely transformed into Z phase. The relevant reasons can be explained as follows.

During the solidification process, some alloy elements such as Nb, Cr, and N are gradually enriched in the residual liquid phase, promoting the formation of (Cr,Nb)N. It can be expected that there exist a strong chemical driving force for phase transformation between (Cr,Nb)N and its surrounding matrix, due to the significant difference in elements (Cr, Mo, Nb, N, etc.) content between them. Therefore, during heat treatment at 1180 °C, the Nb element in (Cr,Nb)N was gradually replaced by the Cr and Mo elements in surrounding matrix, corresponding to the transformation from (Cr,Nb)N to Z phase.

Supplementary Fig. 12 Chemical composition variation of Nb-bearing phase in as-cast 0.25Nb S32205 DSS after heat treatment at 1180 °C for different times.

To sum up, (Cr,Nb)N preferentially precipitated during the solidification process to form inclusion@(Cr,Nb)N core-shell structure. Subsequently, the (Cr,Nb)N phase gradually transformed into Z phase during the heat treatment process before hot working. Accordingly, the inclusion@(Cr,Nb)N core-shell structure transformed into inclusion@Z core-shell structure. Based on these facts, we have revised the corresponding descriptions and discussions in the manuscript. Thanks again for the reviewer’s very insightful and valuable comments.

Revision in the manuscript:

By comparison, the addition of Nb promotes the formation of (Cr,Nb)N whose initial precipitation temperature (1370 °C) is also lower than those of the inclusions^{37,38}. Around 1250 °C, the (Cr,Nb)N phase completely transforms into Z phase which can exist stably during hot working and heat treatment procedures (as indicated by the light blue shaded area in Fig. 1c). Furthermore, the (Cr,Nb)N and Z phases exhibit very similar precipitation and transformation behaviors within the composition range of key elements (Cr, Mo, N, and Nb) of the steel (**Supplementary Fig. 1**). Thus, in the final product, it is very likely to form a structure of Z phase wrapped on inclusion. Interestingly, the Z phase mainly contains approximately 50 wt.% Nb and a certain amount of Cr, Mo, and N (Fig. 1f). In particular, its Cr content is close to that of the steel matrix. This indicates that the Z phase not only is a corrosion-resistant precipitate but also does not induce Cr depletion in the matrix. Therefore, Nb microalloying shows considerable promise in terms of wrapping inclusions with Nb-bearing phase without inducing itself and matrix corrosion. (**Pages 6-7, Lines 107-119**)

(Cr,Nb)N belongs to a NbN-type precipitate, which has a face-centered cubic structure ($a = 0.439$ nm), while Z phase has a tetragonal structure ($a = 0.3037$ nm and $c = 0.7391$ nm). (**Page 7, Lines 122-124**)

During the subsequent solidification process, the ferrite (δ) phase, MnS inclusion, and austenite (γ) phase start to precipitate from the liquid steel at approximately 1444, 1418, and 1305 °C, respectively. Reportedly^{51,52}, MnS is prone to nucleating around the pre-existing MgAl_2O_4 to form MgAl_2O_4 -MnS composite inclusion. As solidification continues, some alloy elements are gradually enriched in the residual liquid phase, leading to apparent positive segregation⁵³. At approximately 1295 °C, the enrichments of Nb, Cr, and N promote the precipitation of (Cr,Nb)N. Because the lattice misregistries between (Cr,Nb)N and the main inclusions (MgAl_2O_4 and MnS) are much lower than the coherency criterion of 12%³⁵ (**Supplementary Table 2**), so (Cr,Nb)N can easily nucleate around these inclusions, thereby forming $\text{MgAl}_2\text{O}_4@(\text{Cr,Nb})\text{N}$, $\text{MnS}@(\text{Cr,Nb})\text{N}$, and $\text{MgAl}_2\text{O}_4\text{-MnS}@(\text{Cr,Nb})\text{N}$ core-shell structures (**Supplementary Fig. 11**). (**Pages 13-14, Lines 271-281**)

Before hot working, the cast ingot experienced a heat treatment at 1180 °C for 1 h. Interestingly, the (Cr,Nb)N phase completely transformed into Z phase during the heating process (**Supplementary Fig. 12**). This phase transformation is mainly achieved through the continuous replacement of Nb in the (Cr,Nb)N by Cr and Mo in the surrounding matrix, due to the strong chemical driving force between them. Accordingly, the inclusion@ (Cr,Nb)N core-shell structure also transformed into inclusion@Z core-shell structure. (**Page 14, Lines 282-290**)

Comment 4: In Supp. Fig.6 the argument with Z phase being able to wrap around MnS phase does not hold here in this case, with MnS precipitating at lower temperatures compared to the apparent Z-phase here. The primary inclusions are present (Mg-Al-O), with the ferrite and austenite then forming at lower temperatures, followed by “Z-phase” formation.

Response: We thank the reviewer for the very pertinent comment. As mentioned in the previous reply, we recalculated the nonequilibrium solidification process using FactSage software. From the new calculation result (**Supplementary Fig. 10**), it can be found that MnS formed after primary MgAl_2O_4 , with (Cr,Nb)N forming at lower temperature. Thus, the (Cr,Nb)N phase can wrap MgAl_2O_4 and/or MnS inclusions, which has been verified by the SEM microstructure observations (**Supplementary Fig. 11**). The relevant descriptions and discussions have been revised in the manuscript. (**Pages 13-14, Lines 271-281**)

Comment 5: Nb is a ferrite former, with also a secondary effect to bind to nitrogen, taking N away from austenite formers. N is also believed to be responsible for the corrosion resistance of the austenite phases. This will negatively affect the “equilibrium” phase ratio here, but again this is not discussed at all. The phase ratio is reported as 50:50, which might be related to non-equilibrium cooling conditions.

Response: Thanks for the reviewer's very insightful suggestions. We fully agree that Nb is a ferrite former and can take N away from austenite, which will affect the equilibrium phase ratio. Inspired by this comment, we realized that it is not rigorous to evaluate the phase ratio of each steel using only one image. Therefore, we remeasured the phase ratios of the three steels using at least 30 metallographic micrographs (**Supplementary Fig. 3d**). The results show that Nb addition indeed leads to a slight increase in the ferrite phase fraction, which meets the above expectations. As mentioned by the reviewer, when Nb takes a few N away from austenite, it is theoretically detrimental to corrosion resistance. But we observe that the corrosion resistance of Nb-bearing steels has noticeably improved (**Fig. 3**). This further indicates that the inclusions wrapped by Z phase plays an absolutely dominant role in significantly improving corrosion resistance of the steel.

Supplementary Fig. 3 Characterization of heat-treated microstructures. Typical microstructure and ferrite phase fractions of S32205 DSSs solution treated at 1050 °C for 0.5 h: **a** 0Nb, **b** 0.10Nb, **c** 0.25Nb, and **d** ferrite phase fractions.

Revision in the manuscript:

Additionally, the three steels after hot working and heat treatment have ferrite-austenite duplex microstructures, and the addition of Nb leads to a slight increase in the ferrite phase fraction because Nb is a ferrite former and can take a few N away from austenite (**Supplementary Fig. 3**). (Page 7, Lines 136-139)

Comment 6: (iii) Corrosion - Fig.3 (a) indicates better pitting corrosion behaviour for pit growth (higher breakaway potential), but similar pit nucleation potentials for meta-stable pits. All materials seem to show metastable pits forming at around +200 to +300 mV above the rest potential (E_{Corr}) in these test conditions, with stable pits then more likely growing in samples without Nb additions. But current spikes of Nb containing microstructures are clearly visible in Fig.3a at these low potentials. It looks like that pits are still initiated, but far fewer in number for Nb containing microstructures, reducing the probability to transition to stable pitting. This observation is interesting.

Response: We thank the reviewer a lot for his/her meticulous observation and also for recognizing the interest of the electrochemical corrosion results. Inspired by this comment, we further observed the corrosion morphologies of inclusions and inclusion@Z core-shell structure after potentiodynamic polarization tests stopped at 100 mV, 200 mV, and pitting potential (Fig. 3b and c). At 100 mV and 200 mV, visible metastable pits formed around the individual inclusions in 0Nb steel and further expanded into large-size corrosion pits at the pitting potential. For the inclusion@Z core-shell structure in 0.25Nb steel, although metastable pits also formed around the core inclusion at low potential, they did not expand significantly at the pitting potential. These phenomena indicate that the Z phase is indeed very corrosion resistant, so that local acidification in the pit caused by inclusion dissolution will not induce apparent corrosion of Z phase. Meanwhile, the corrosion-resistant Z phase shell effectively prevents the development of metastable pit into stable pit.

Fig. 3 Electrochemical and immersion corrosion behaviour of S32205 DSSs. Electrochemical corrosion behaviour in double-concentration simulated seawater at 72 °C (pH 8.2): **a** potentiodynamic polarization curves and corrosion morphologies of **b** 0Nb and **c** 0.25Nb steels.

Revision in the manuscript:

Fig. 3b and c show the corrosion morphologies of inclusions and inclusion@Z core-shell structure after potentiodynamic polarization tests stopped at 100 mV, 200 mV, and pitting potential. At 100 mV and 200 mV, visible metastable pits formed around the individual inclusions in 0Nb steel and further expanded into large-size corrosion pits at the pitting potential. For the inclusion@Z core-shell structure in 0.25Nb steel, although metastable pits also formed around the core inclusion at low potential, they did not expand significantly at the pitting potential. These phenomena indicate that the Z phase is indeed very corrosion resistant, so that local acidification in the pit caused by inclusion dissolution will not induce apparent corrosion of Z phase. Meanwhile, the corrosion-resistant Z phase shell effectively prevents the development of metastable pit into stable pit. (Pages 9-10, Lines 180-190)

Comment 7: Figs. 3 & 5 clearly show that the cathodic and anodic branches of the electrochemical tests are nearly superimposed on each other, yielding the same passive current density and (most likely) very similar i_{corr} values (just be visual approximation). The increase in corrosion rate in Fig. 3c is then only accessible if localised corrosion is considered here (which is most likely the case for FeCl_3 exposure). Also, I would not expect for Nb to play a role in the passive film composition with such a small concentration and characteristic of atom.

Response: Thanks for the reviewer's insightful comments. For the electrochemical tests, the cathodic reaction is the oxygen reduction, and the anodic reaction is the anodic dissolution of metallic atoms M into ions M^{n+} . Based on the Thermo-Calc calculation results (Fig. R1), the addition of minor Nb has a slight effect on the chemical compositions of the steel matrix, due to only a few precipitations of Nb-bearing phases. In other words, the steel matrix compositions of the three DSSs are basically the same. Thus, the cathodic and anodic reactions of the DSSs in the Tafel regions are also the same, which undoubtedly contributed to the superimposed cathodic and anodic branches of the potentiodynamic polarization curves, as well as the almost same corrosion current density (i_{corr}) and corrosion potential (E_{corr}). Besides, the XPS results (Supplementary Fig. 7) indicate that the composition of passive films on the three DSSs are basically the same. Therefore, the passive current densities of these DSSs are also basically the same.

However, during the immersion corrosion tests in 6% FeCl_3 solution, the three DSSs exhibit different response. For the 0Nb steel, pitting corrosion preferentially initiated around inclusions, and grew into the steel matrix, thus inducing a significant weight loss (i.e., high corrosion rate). While for the Nb-bearing steels, especially the 0.25Nb steel, the corrosion-

resistant Z phase can wrap the deleterious inclusions and effectively restrain the development of pitting corrosion to the steel matrix, thereby significantly reducing the corrosion rate.

Fig. R1 Effect of Nb content on elemental compositions of **a** ferrite (δ) and **b** austenite (γ) phases.

Comment 8: I am really surprised that just one “standard composition” was used for the ThermoCalc simulations (Suppl.tab1 & tab.5) rather than the exact chemical compositions. For using the standard composition, I would expect to use the full range (min. – to max.) compositions of key elements; some of the values used may significantly affect the overall precipitation behaviour.

Response: We thank the reviewer for the very pertinent comment. After fully considering the suggestions, we have conducted supplementary calculations in the following aspects. In the section of “Microalloying element selection”, we first selected Nb as the ideal microalloying element for our strategy through Thermo-Calc calculations using standard compositions (**Supplementary Table 1**). To further verify the applicability of Nb microalloying, we calculated the precipitation behaviour of S32205 DSS using the full range (min. to max.) compositions of key elements (Cr, Mo, N, and Nb), as shown in **Supplementary Fig. 1**. The results show that both (Cr,Nb)N and Z phases exhibit very similar precipitation and transformation behaviors within the composition range of key elements of S32205 DSS, indicating a broad applicability of our strategy.

In the section of “Universality of Nb microalloying technology”, we recalculated the precipitation behaviour of various microalloyed DSSs (including lean S32101 and S32304, standard S32205, super S32750, and hyper S32707) using exact chemical compositions (**Supplementary Table 5**). The new results (**Fig. 5a**) indicate that Nb microalloying is also a widely applicable strategy for these DSSs.

Supplementary Table 1 Chemical compositions (wt.%) of microalloyed S32205 DSSs used for Thermo-Calc calculations.

Steels	C	Si	Mn	Cr	Ni	Mo	N	Ti	V	Nb	Fe
Ti0.25	0.02	0.5	1.2	22.5	5.5	3.0	0.16	0.25	–	–	Bal.
V0.25	0.02	0.5	1.2	22.5	5.5	3.0	0.16	–	0.25	–	Bal.
Nb0.25	0.02	0.5	1.2	22.5	5.5	3.0	0.16	–	–	0.25	Bal.
Cr22.0	0.02	0.5	1.2	22.0	5.5	3.0	0.16	–	–	0.25	Bal.
Cr23.0	0.02	0.5	1.2	23.0	5.5	3.0	0.16	–	–	0.25	Bal.
Mo3.2	0.02	0.5	1.2	22.5	5.5	3.2	0.16	–	–	0.25	Bal.
Mo3.5	0.02	0.5	1.2	22.5	5.5	3.5	0.16	–	–	0.25	Bal.
N0.14	0.02	0.5	1.2	22.5	5.5	3.0	0.14	–	–	0.25	Bal.
N0.20	0.02	0.5	1.2	22.5	5.5	3.0	0.20	–	–	0.25	Bal.
Nb0.15	0.02	0.5	1.2	22.5	5.5	3.0	0.16	–	–	0.15	Bal.
Nb0.35	0.02	0.5	1.2	22.5	5.5	3.0	0.16	–	–	0.35	Bal.

Note: The chemical compositions (wt.%) of Cr22.5, Mo3.0, and N0.16 are same with that of Nb0.25.

Supplementary Fig. 1 Effect of key elements content on the precipitation behavior of Nb-bearing phases in S32205 DSSs: **a** Cr, **b** Mo, **c** N, and **d** Nb.

Supplementary Table 5 Chemical compositions (wt.%) of DSSs used for Thermo-Calc calculations.

Types	Steels	C	Si	Mn	Cr	Ni	Mo	Cu	Co	N	Nb	Fe
Lean	S32101	0.016	0.58	5.02	21.50	1.51	0.32	0.49	–	0.22	0.25	Bal.
	S32304	0.015	0.53	1.46	23.12	4.55	0.33	0.44	–	0.12	0.25	Bal.

Standard	S32205	0.015	0.50	1.20	22.52	5.59	3.13	–	–	0.16	0.25	Bal.
Super	S32750	0.016	0.52	0.71	25.05	7.02	4.01	–	–	0.28	0.25	Bal.
Hyper	S32707	0.014	0.48	1.12	26.86	7.08	4.63	0.95	0.97	0.41	0.25	Bal.

Fig. 5 Universality of Nb microalloying technology. a Precipitation behaviour of Nb-bearing phases in series of DSSs microalloyed with 0.25 wt.% Nb, which were calculated by Thermo-Calc software.

Revision in the manuscript:

Furthermore, the (Cr,Nb)N and Z phases exhibit very similar precipitation and transformation behaviors within the composition range of key elements (Cr, Mo, N, and Nb) of the steel (**Supplementary Fig. 1**). (**Page 6, Lines 111-113**)

The results show that Z phase can form in all types of Nb microalloyed DSSs through (Cr,Nb)N transformation or direct precipitation, and their initial formation temperatures (1318–1378 °C) are rather lower than those of inclusions^{37,38}. Accordingly, the Z phase is also greatly promising for wrapping inclusions in these DSSs. (**Page 12, Lines 237-240**)

Comment 9: The work reminded me on some of the earlier work on precipitation strengthened Ni-base alloys 718 where different (Al+Ti)/Nb -alloy ratios were used to create sandwich morphologies of the strengthening phases (γ'') and (γ'). This was back then an interesting approach, that was unfortunately not developed further (to my knowledge). [5]

[5] Cozar, R. & Pineau, A.: Morphology of γ' and γ'' Precipitates and Thermal Stability of Inconel 718 Type Alloys. Metallurgical Transactions A, Vol. 4, No. 1, p.47-59, 1973.

Response: We highly appreciate the reviewer for his/her broad knowledges about the earlier interesting work on creating sandwich morphologies of the strengthening γ' and γ'' phases in Inconel 718 alloys with different (Al+Ti)/Nb ratios.

REVIEWER COMMENTS

Reviewer #3 (Remarks to the Author):

Anyway, the authors have put a lot of additional work & effort into the manuscript and addressed my concerns. Additional data is included to support their approach and statements, supported by FactSage modelling. There is still a bit of uncertainty of the universal applicability of this approach, but this is (& will be) always part of research and a healthy bit of uncertainty is acceptable. nous

I would have liked a bit more work around the e-chem behaviour of the Z-phase, NbC, corshell phase in aqueous environment; but the SKPFM proxy measurements are acceptable (even though these might be affected by whatever environment you do your scans in).

Most of my initial concerns have been addressed & discussed in the rebuttal letter. The manuscript still contains some odd phrases/expressions and a good proof read is certainly required here.

Reviewer #4 (Remarks to the Author):

Dear authors,

I have been asked to review, whether all comments from reviewer #2 have been properly considered by you.

Key results:

The approach of the authors is very innovative and creative and I like the results. The authors:

- alloy a Duplex Stainless Steel (DSS) with low amounts of Nb (up to 0.25Nb) and a layer of Z phase is precipitated as shell at a high fraction (75%) around MnS-MgAl₂O₄ nonmetallic inclusions.
- the Z phase shell consists of (ca.) 55Nb-27Cr-5Mo-5Fe-8N
- this results in an improvement of corrosion properties since nonmetallic inclusions, especially MnS, decrease especially pitting resistance.
- Cr content of the Z phase is equal to Cr content of the DSS matrix and therefore is no Cr depletion and no decrease of corrosion properties of the matrix.

Comment 1 of Reviewer #2:

This is just a remark and had not to be considered.

Comment 2 of Reviewer #2:

The authors have very seriously and in a complete way answered to the reviewers comment. Supplementary Fig. 6 and the revision in the manuscript are fully satisfactory from my side and I have nothing to add. From my point of view the answer of the authors exceeds even the concerns of the reviewer by doing the Gumpel extreme value analysis.

Comment 3 from Reviewer #2:

The reviewer has concerns that there is definitely a strong positive effect on corrosion properties. He asks for electrochemical reverse scans to determine the repassivation behaviour and for a durability test for the Z-phase. He also asks about the effect of acidification during (mainly) MnS dissolution on the Z phase.

The authors have very properly additionally investigated the durability of the Z phase with long term immersion tests in highly aggressive reducing FeCl₃ solution. They could prove the a high pitting resistance of the Z-phase, which is very understandable due to its composition.

The authors did not do the measurement of the repassivation behaviour and the authors let the question open, because they write that "the pits inside the shell could NOT repassivate during backward scan but did not expand into large size pits". This statement is contradictory (Reviewer report page 8, lines 2-5). So the concern of reviewer 2 is not fully considered.

From my side there is an additional concern: The Z phase is significantly enriched in Mo and N and this enrichment comes from the DSS matrix and necessarily has to generate a Mo and N depletion. Consequently there has to occur a certain reduction of pitting resistance in the vicinity of the Z phase. Fig. 4 d shows some evidence for this (large amount of pits close to Z-phase), while Supplementary Fig. 9 does not show evidence for this.

My last concern regarding this point is the statement of the authors:
page 6, last line of the answer to the reviewer report: "We can firmly confirm that Z phase ... can effectively prevent corrosion caused by inclusions WITHOUT INDUCING OTHER TYPES OF CORROSION."

The authors have investigated the pitting behaviour but claim the good results for other types of corrosion as well. Enriched precipitates like Z phase shell formation surrounding MnS-MgAl₂O₄ can have a beneficial effect on pitting. On the other side any enrichment of Mo or N in a phase yields to a depletion in the surrounding matrix of this phase and can cause substantial decrease of corrosion properties such as resistance to intergranular corrosion.

Therefore this general claim of the authors is to my opinion not justified and I follow the recommendation of Reviewer #2 to submit the paper as a metallurgical paper (the shelling effect of Z phase is great and beautiful, congratulations for the finding!) and to elaborate more deeply on corrosion properties. There is no gain in corrosion properties due to an enrichment of beneficial elements in a phase (Z phase) without a depletion of the surrounding accompanied with a (at least certain) loss of corrosion properties. This loss of corrosion properties has to be shown first and its extent has to be quantified or a very good argumentation has to be found, why it is definitely not there.

Comment 4 from Reviewer #2:

The reviewer repeats his concerns about corrosion properties that they are not complete. The authors answer that there is no Cr depletion due to similar composition of Cr in the DSS matrix and the Z phase. The enrichment of Mo and N remains unmentioned (see also above). Mo is enriched in the passive layer and therefore can be depleted in the vicinity of the Z phase and N is enriched underneath the passive layer favouring repassivation and therefore can also be depleted in the vicinity of the Z phase and repassivation can be hindered. That underlines the missing repassivation measurement from the Reviewer #2 in the Comment 3.

Comment 5 from Reviewer #2:

I follow the proposal of Reviewer #2 to go for a metallurgical paper and to more deeply evaluate corrosion behaviour in a separate work. See Comments 3 and 4 above.

Comment 6 from Reviewer #2:

The additional comment on galvanic corrosion is answered and revised sufficiently by the authors.

Point-by-point response to reviewers' comments (NCOMMS-22-44107B)

We sincerely thank the reviewers for carefully reading our revised manuscript and for the additional constructive comments and suggestions which significantly strengthen the presentation and impact of our work. Detailed point-by-point response to the reviewers' comments and suggestions are summarized below. We have carefully revised the manuscript and supplementary materials, and all changes are marked in yellow.

Reviewer #3

Comment 1: Anyway, the authors have put a lot of additional work & effort into the manuscript and addressed my concerns. Additional data is included to support their approach and statements, supported by FactSage modelling. There is still a bit of uncertainty of the universal applicability of this approach, but this is (& will be) always part of research and a healthy bit of uncertainty is acceptable. Nous

Response: We highly appreciate the reviewer's recognition of our additional work. We would like to provide a few more explanations on using the FactSage software to calculate the nonequilibrium solidification process of 0.25Nb S32205 DSS. Firstly, we used the latest database version (8.2), which was constructed based on a large amount of experimental data from existing literatures. Moreover, the solidification process of steel is actually nonequilibrium, and we did use the Gulliver-Scheil model in FactSage software to calculate this nonequilibrium solidification process, and the calculation result was well consistent with our experimental results. There might be still some room to further improve this modelling, but we believe that our approach is scientific and reasonable.

Comment 2: I would have liked a bit more work around the e-chem behaviour of the Z-phase, NbC, corshell phase in aqueous environment; but the SKPFM proxy measurements are acceptable (even though these might be affected by whatever environment you do your scans in).

Response: Thanks for the reviewer's very insightful comments. It is indeed important to pay attention to the electrochemical behaviour of the Z phase and core-shell structure in aqueous environment. However, because the Z phase and the core-shell structure are only around 1–2 microns, it is difficult to accurately measure the electrochemical behavior of precipitates with such a small size using micro-electrochemical method such as scanning vibrating electrode

technique (SVET). Therefore, we adopted the SKPFM method with higher testing accuracy to measure the potential difference between the Z phase and its surrounding matrix to evaluate the possibility of galvanic corrosion between them. Finally, we thank the reviewer a lot for recognizing our SKPFM measurement.

Comment 3: Most of my initial concerns have been addressed & discussed in the rebuttal letter. The manuscript still contains some odd phrases/expressions and a good proof read is certainly required here.

Response: We are happy that most of the reviewer's initial concerns have been resolved. We greatly appreciate the reviewer's suggestion on proofing the phrases/expressions. We have double-checked the phrases/expressions and polished the language throughout the manuscript. In addition, the revised manuscript has been sent to a professional institution (AJE) to polish carefully, and the revised portions are marked in red.

Finally, we would like to thank the reviewer once again for providing such valuable comments and constructive suggestions which are important for improving the integrity and impact of our work.

Reviewer #4

Dear authors,

I have been asked to review, whether all comments from reviewer #2 have been properly considered by you.

Key results:

The approach of the authors is very innovative and creative and I like the results. The authors:

- alloy a Duplex Stainless Steel (DSS) with low amounts of Nb (up to 0.25Nb) and a layer of Z phase is precipitated as shell at a high fraction (75%) around MnS-MgAl₂O₄ nonmetallic inclusions.

- the Z phase shell consists of (ca.) 55Nb-27Cr-5Mo-5Fe-8N

- this results in an improvement of corrosion properties since nonmetallic inclusions, especially MnS, decrease especially pitting resistance.

- Cr content of the Z phase is equal to Cr content of the DSS matrix and therefore is no Cr depletion and no decrease of corrosion properties of the matrix.

Response: We thank the reviewer a lot for reviewing our revised manuscript. We also greatly appreciate the reviewer for high recognition of the innovation and impact of our work.

Comment 1 of Reviewer #2:

This is just a remark and had not to be considered.

Response: Thank you.

Comment 2 of Reviewer #2:

The authors have very seriously and in a complete way answered to the reviewers comment. Supplementary Fig. 6 and the revision in the manuscript are fully satisfactory from my side and I have nothing to add. From my point of view the answer of the authors exceeds even the concerns of the reviewer by doing the Gumpel extreme value analysis.

Response: We greatly appreciate the reviewer for high recognition of our responses to this comment.

Comment 3 from Reviewer #2: (New comment 1)

The reviewer has concerns that there is definitely a strong positive effect on corrosion properties. He asks for electrochemical reverse scans to determine the repassivation behaviour and for a durability test for the Z-phase. He also asks about the effect of acidification during (mainly) MnS dissolution on the Z phase.

The authors have very properly additionally investigated the durability of the Z phase with long term immersion tests in highly aggressive reducing FeCl_3 solution. They could prove the a high pitting resistance of the Z-phase, which is very understandable due to its composition.

The authors did not do the measurement of the repassivation behaviour and the authors let the question open, because they write that "the pits inside the shell could NOT repassivate during backward scan but did not expand into large size pits". This statement is contradictory (Reviewer report page 8, lines 2-5). So the concern of reviewer 2 is not fully considered.

Response: Thanks for the reviewer's very valuable suggestion, and we apologize for the inaccurate statement. We conducted the cyclic polarization tests (**Supplementary Fig. 7a**) in double-concentration simulated seawater at 72 °C (pH 8.2). It can be found that the protective potentials of Nb-bearing steels are very close to that of Nb-free steel, indicating that these steels exhibited almost the same repassivation behaviours. According to the corrosion morphologies of samples after potentiodynamic polarization (**Fig. 3c**) and immersion corrosion tests (**Fig. 4**), the corrosion resistance of the Z phase is higher than the matrix and much higher than the inclusions. This means that during the forward scan of cyclic polarization, corrosion in the Nb-bearing steel usually occurred at the steel matrix with defects rather than the Z phase. Then, during the reverse scan, the repassivation also occurred at the front of corroded steel matrix. Therefore, both Nb-bearing and Nb-free steels underwent repassivation in the steel matrix (**Supplementary Fig. 7b**), so they exhibited similar repassivation behaviours.

In summary, it is not possible to measure the repassivation behaviour of the Z phase through cyclic polarization method, because the Z phase has not been corroded. In addition, the Z phase is only around 1–2 microns, so it is also scarcely possible to measure the repassivation behaviour of precipitate with such a small size using micro-electrochemical methods.

Supplementary Fig. 7 Repassivation behaviour of S32205 DSSs. **a** Cyclic polarization curves in double-concentration simulated seawater at 72 °C (pH 8.2), and **b** Schematic diagram of corrosion and repassivation process.

Revision in the manuscript:

Supplementary Fig. 7a shows the cyclic polarization curves in simulated seawater, which indicate that the Nb-bearing steels exhibited very similar protective potentials (repassivation ability) to the Nb-free steel. As schematically shown in **Supplementary Fig. 7b**, neither the Z phase nor its surrounding matrix in Nb-bearing steel was corroded. Thus, corrosion mainly occurred at the steel matrix with defects. Accordingly, the repassivation also occurred at the front of the corroded steel matrix. Therefore, both Nb-bearing and Nb-free steels underwent repassivation in the steel matrix, so they exhibited similar repassivation behaviours. **(Page 10, Lines 196-203)**

Comment 3 from Reviewer #2: (New comment 2)

From my side there is an additional concern: The Z phase is significantly enriched in Mo and N and this enrichment comes from the DSS matrix and necessarily has to generate a Mo and N depletion. Consequently there has to occur a certain reduction of pitting resistance in the vicinity of the Z phase. Fig. 4 d shows some evidence for this (large amount of pits close to Z-phase), while Supplementary Fig. 9 does not show evidence for this.

Response: Thanks for the reviewer's very insightful comment. To detect the concentration distribution of elements around the Z phase, we performed line profile analysis across the interface of Z phase and matrix using an aberration-corrected STEM with an associated EDS (**Supplementary Fig. 5**). As expected, slightly Mo- and N-depleted zones were detected near the Z phase. Theoretically, the formation of depleted zones should reduce the corrosion resistance to a certain extent. However, after potentiodynamic polarization tests in simulated seawater, there was no sign of corrosion in the regions around the Z phase (**Fig. 3c**). This indicates that the Mo- and N-depleted zones still had good corrosion resistance in this environment. This is because the depleted zones still contained relatively high contents of Cr, Mo, and N elements (**Supplementary Fig. 5**). As pointed out by the reviewer, large amounts of pits formed close to the Z phase after long-term immersion tests in highly aggressive FeCl_3 solution (**Fig. 4d**). This reveals that the Mo- and N-depleted zones were indeed corroded in extremely harsh environments. However, we should note that the steel matrix far from the depleted zones also underwent severe corrosion (**Fig. R1** and **Supplementary Fig. 11**). In other words, in such a harsh environment, the steel matrix has already undergone corrosion, the corrosion of the depleted zones should also be acceptable.

Supplementary Fig. 5 STEM and EDS characterization of the Z phase. a STEM dark-field image of the Z phase, and **b** STEM-EDS line profile analysis across the Z phase.

Fig. R1 SEM morphology of 0.25Nb steel after immersion corrosion in 6% FeCl₃ solution at 50 °C for 10 d. (Low magnification of Fig. 4d)

Revision in the manuscript:

Additionally, no Cr-depleted zone and only slightly Mo- and N-depleted zones were detected near the Z phase (**Supplementary Fig. 5**), and their effect on the corrosion resistance will be discussed in the next section. (**Page 8, Lines 150-152**)

Meanwhile, the matrix surrounding the Z phase was also not corroded after potentiodynamic polarization testes, indicating that it still had good corrosion resistance in simulated seawater. (**Page 10, Lines 194-196**)

Notably, in highly aggressive 6% FeCl₃ solution, the steel matrix far from the Z phase has already corroded, so the corrosion of the regions surrounding the Z phase should be acceptable. (**Page 12, Lines 241-243**)

Moreover, although slightly Mo- and N-depleted zones were detected near the Z phase, these regions still contained relatively high contents of Cr, Mo, and N elements (**Supplementary Fig. 5**), so they still had good corrosion resistance in simulated seawater. (**Pages 16-17, Lines 338-341**)

Although parts of the matrix surrounding the Z phase were corroded after long-term immersion in highly aggressive 6% FeCl₃ solution, the steel matrix far from the Z phase was also corroded (Fig. 4d and **Supplementary Fig. 11**). Therefore, this universal corrosion in extremely harsh environments is acceptable. (**Page 17, Lines 357-360**)

Comment 3 from Reviewer #2: (New comment 3)

My last concern regarding this point is the statement of the authors:

page 6, last line of the answer to the reviewer report: "We can firmly confirm that Z phase ... can effectively prevent corrosion caused by inclusions WITHOUT INDUCING OTHER TYPES OF CORROSION."

The authors have investigated the pitting behaviour but claim the good results for other types of corrosion as well. Enriched precipitates like Z phase shell formation surrounding MnS-MgAl₂O₄ can have a beneficial effect on pitting. On the other side any enrichment of Mo or N in a phase yields to a depletion in the surrounding matrix of this phase and can cause substantial decrease of corrosion properties such as resistance to intergranular corrosion.

Response: We greatly appreciate the reviewer's rigorous comments and suggestions, and we apologize for the imprecise expression. To clarify the effect of Mo and N depletion around the Z phase on intergranular corrosion (IGC) resistance, we performed double loop electrochemical potentiokinetic reactivation (DL-EPR) tests in a 2 M H₂SO₄ + 0.75 M HCl + 0.01 M KSCN solution. **Supplementary Fig. 12** shows that the degree of sensitization ($R = (I_f/I_a) \times 100$) of Nb-bearing steels are basically the same with that of the Nb-free steel, indicating that the slight Mo and N depletion surrounding the Z phase has a negligible effect on the IGC resistance of the Nb-bearing steels. To clarify the relevant reasons, we observed the IGC morphologies (**Supplementary Fig. 13**) of samples after the DL-EPR tests. Apparently, corrosion mainly occurred within the γ phase and at the γ/δ boundaries. Although the depleted zones around the Z phase were also corroded, this corrosion was negligible compared to the corrosion of the γ phase and the γ/δ boundaries. This is because the total area of the depleted zones around the Z phases is much smaller than that of the γ phase and the γ/δ boundaries. Therefore, it can be concluded that our strategy of wrapping deleterious inclusions with corrosion-resistant niobium armour (Z phase) has a negligible influence on the IGC resistance of DSSs.

Supplementary Fig. 12 Double loop electrochemical potentiokinetic reactivation (DL-EPR) results of S32205 DSSs. a Typical DL-EPR curves, and **b** Degree of sensitization (R value). $R=(I_r/I_a)\times 100$, I_a and I_r represent the peak activation current density and peak reactivation current density measured during the forward and reverse scans, respectively.

Supplementary Fig. 13 IGC morphologies of S32205 DSSs after DL-EPR tests. a 0Nb, **b** 0.25Nb, **c** and **d** Enlarged views of the regions circled by the dashed boxes in (b).

Revision in the manuscript:

In addition, the double loop electrochemical potentiokinetic reactivation (DL-EPR) results (Supplementary Fig. 12) show that the Nb-bearing steels exhibit similar degrees of sensitization to the Nb-free steel, indicating that the slight Mo and N depletion surrounding the Z phase has a negligible effect on the intergranular corrosion (IGC) resistance of the Nb-bearing

steels. Apparently, corrosion mainly occurred within the γ phase and at the γ/δ boundaries (**Supplementary Fig. 13**). Although the depleted zones around the Z phase were also corroded, this corrosion was negligible compared to the extensive corrosion of the γ phase and the γ/δ boundaries because the total area of the depleted zones was much smaller than that of the γ phase and the γ/δ boundaries. (**Pages 12-13, Lines 244-252**)

Additionally, the slight Mo and N depletion around the Z phase has a negligible influence on the repassivation behaviour (**Supplementary Fig. 7**) and IGC resistance of the steel (**Supplementary Figs. 12 and 13**). (**Pages 17-18, Lines 360-362**)

Comment 3 from Reviewer #2: (New comment 4)

Therefore this general claim of the authors is to my opinion not justified and I follow the recommendation of Reviewer #2 to submit the paper as a metallurgical paper (the shelling effect of Z phase is great and beautiful, congratulations for the finding!) and to elaborate more deeply on corrosion properties. There is no gain in corrosion properties due to an enrichment of beneficial elements in a phase (Z phase) without a depletion of the surrounding accompanied with a (at least certain) loss of corrosion properties. This loss of corrosion properties has to be shown first and its extent has to be quantified or a very good argumentation has to be found, why it is definitely not there.

Response: We thank the reviewer for the very pertinent comment and suggestion. We apologize for not providing sufficient experimental data and discussion to support our conclusions in the last revised manuscript. As replied above, we have carried out further STEM-EDS analysis, DL-EPR tests, and microstructure characterization to confirm that the Mo and N depletion surrounding the Z phase has a negligible effect on the loss of corrosion properties. Here, we summarize all our corrosion results as follows:

(1) Potentiodynamic polarization and cyclic polarization tests in double-concentration simulated seawater at 72 °C (pH 8.2)

- The addition of 0.25 wt.% Nb doubly enhanced the pitting potential of S32205 DSSs. (**Fig. 3a** and **Supplementary Fig. 7a**)

- After the tests, the Z phase and its surrounding matrix were not corroded. (**Fig. 3c**)

- Nb-bearing steels exhibited almost the same repassivation behaviour to the Nb-free steel. (**Supplementary Fig. 7a**)

(2) Immersion corrosion tests in 6% FeCl₃ solution at 50 °C

- The addition of Nb led to remarkable reductions in the corrosion rate as well as the quantity, maximum diameter, and maximum depth of the pit cavities. (**Fig. 3d–g**)

- After immersion for 10 d, the Z phase was not corroded, and the corrosion-resistant Z phase shell effectively prevented the galvanic or microcrevice corrosion caused by inclusions. (Fig. 4d)

- After immersion for 10 d, the steel matrix has already corroded, so the corrosion of the depleted zones surrounding the Z phase should also be acceptable. (Fig. 4d and **Supplementary Fig. 11**)

(3) DL-EPR tests in a 2 M H₂SO₄ + 0.75 M HCl + 0.01 M KSCN solution at 30 °C with a scan rate of 1 mV·s⁻¹

- Nb-bearing steels exhibited almost the same IGC resistance to the Nb-free steel. (**Supplementary Fig. 12**)

- The slight corrosion of the depleted zones around the Z phase can be negligible compared to the extensive corrosion of the γ phase and the γ/δ boundaries. (**Supplementary Fig. 13**)

- The slight Mo and N depletion surrounding the Z phase has a negligible effect on the IGC of the Nb-bearing steels. (**Supplementary Figs. 12 and 13**)

In summary, our strategy of wrapping deleterious inclusions with corrosion-resistant niobium armour (Z phase) indeed effectively prevents corrosion caused by inclusions. Although the regions around the Z phase may be corroded in extremely harsh environments, it appears insignificant compared to the extensive corrosion of the steel matrix, and therefore is acceptable. Overall, our strategy can significantly improve the corrosion resistance of DSSs.

Comment 4 from Reviewer #2:

The reviewer repeats his concerns about corrosion properties that they are not complete. The authors answer that there is no Cr depletion due to similar composition of Cr in the DSS matrix and the Z phase. The enrichment of Mo and N remains unmentioned (see also above). Mo is enriched in the passive layer and therefore can be depleted in the vicinity of the Z phase and N is enriched underneath the passive layer favouring repassivation and therefore can also be depleted in the vicinity of the Z phase and repassivation can be hindered. That underlines the missing repassivation measurement from the Reviewer #2 in the Comment 3.

Response: Thanks for the reviewer's very insightful comment again. As mentioned above, although the slightly Mo- and N-depleted zones were detected around the Z phase, they still contained relatively high contents of Cr, Mo, and N elements (**Supplementary Fig. 5**). Accordingly, the depleted zones still had good corrosion resistance, and they were not corroded after potentiodynamic polarization tests (Fig. 3c). Thus, during the forward scan of cyclic polarization, corrosion in Nb-bearing steel usually occurred at the steel matrix with defects

rather than the Z phase and the surrounding depleted zones (**Supplementary Fig. 7b**). Then, during the reverse scan, the repassivation also occurred at the front of corroded steel matrix with defects. Therefore, the slight Mo and N depletion did not affect the repassivation behaviour of the Nb-bearing steels due to the good corrosion resistance of the depleted zones in simulated seawater (**Supplementary Fig. 7a**). Overall, wrapping deleterious inclusions with the Z phase significantly improves the corrosion resistance of the Nb-bearing steels.

Supplementary Fig. 7 Repassivation behaviour of S32205 DSSs. a Cyclic polarization curves in double-concentration simulated seawater at 72 °C (pH 8.2), and **b** Schematic diagram of corrosion and repassivation process.

Revision in the manuscript:

Supplementary Fig. 7a shows the cyclic polarization curves in simulated seawater, which indicate that the Nb-bearing steels exhibited very similar protective potentials (repassivation ability) to the Nb-free steel. As schematically shown in **Supplementary Fig. 7b**, neither the Z phase nor its surrounding matrix in Nb-bearing steel was corroded. Thus, corrosion mainly occurred at the steel matrix with defects. Accordingly, the repassivation also occurred at the front of the corroded steel matrix. Therefore, both Nb-bearing and Nb-free steels underwent repassivation in the steel matrix, so they exhibited similar repassivation behaviours. (**Page 10, Lines 196-203**)

Additionally, the slight Mo and N depletion around the Z phase has a negligible influence on the repassivation behaviour (**Supplementary Fig. 7**) and IGC resistance of the steel (**Supplementary Figs. 12 and 13**). (**Pages 17-18, Lines 360-362**)

Comment 5 from Reviewer #2:

I follow the proposal of Reviewer #2 to go for a metallurgical paper and to more deeply evaluate corrosion behaviour in a separate work. See Comments 3 and 4 above.

Response: Again, we highly appreciate the reviewer for all above valuable comments and constructive suggestions which are important for improving the integrity and impact of our work. Accordingly, we have strengthened the corrosion results and fully confirmed the credibility of our strategy, as replied above. We have revised the manuscript in full accordance with the reviewer's comments and suggestions. We hope the revision is satisfactory and is now suitable for publication in Nature Communications.

Comment 6 from Reviewer #2:

The additional comment on galvanic corrosion is answered and revised sufficiently by the authors.

Response: We greatly appreciate the reviewer for recognition of our response and revision.

REVIEWERS' COMMENTS

Reviewer #4 (Remarks to the Author):

Dear authors,

I got the task to review your efforts on Reviewer #3 and #4 comments:

Reviewer #3

Comment 1 and 3:

You have substantially revised the paper. Your answers to Comment 1 and 3 are fine for me.

Comment 2:

The Reviewer is not 100% fine with the electrochemical characterization of the Z-phase. The authors have a potential linescan done with SKPFM and for this is ok together with all the other informations supplied.

Reviewer #4

Comment 1, 2 and 6:

The authors have sufficiently corrected, where something had to be done (Comment 1 was more a remark and no actions were needed).

Comment 3, 4 and 5:

These comments are all dealing with the Z-Phase and its surrounding and the resulting corrosion properties. The authors have put in additional linescans through the surrounding of the Z-Phase and the Z-phase itself. They show a very low depletion in Mo and N. Consequently this effect on pitting corrosion and also on other types of corrosion can be expected to be very small.

So therefore the authors have sufficiently clarified my concerns and I am fine with the paper.